**Investigation**

# A maternal germline mutator phenotype in a family affected by heritable colorectal cancer

Candice L. Young,[1,2,†] Annabel C. Beichman,[1,†] David Mas Ponte,[1] Shelby L. Hemker,[3] Luke Zhu,[1,4] Jacob O. Kitzman,[3] Brian H. Shirts,[5] Kelley Harris (ID) [1,6,*]

[1]Department of Genome Sciences, University of Washington, 3720 15th Ave NE, Seattle, WA 98195, USA
[2]Department of Molecular and Cellular Biology, University of Washington, 1705 NE Pacific St, Seattle, WA 98195, USA
[3]Department of Human Genetics, University of Michigan, 1241 Catherine St, Ann Arbor, MI 48109, USA
[4]Department of Bioengineering, University of Washington, 3720 15th Ave NE, Seattle, WA 98195, USA
[5]Department of Laboratory Medicine and Pathology, University of Washington, 1959 NE Pacific St, Seattle, WA 98195, USA
[6]Herbold Computational Biology Program, Fred Hutchinson Cancer Center, P.O. Box 19024, Seattle, WA 98109, USA

*Corresponding author: Department of Genome Sciences, University of Washington, 3720 15th Ave NE, Seattle, WA 98195, USA. Email: harriske@uw.edu
[†]These authors contributed equally to this work.

Variation in DNA repair genes can increase cancer risk by elevating the rate of oncogenic mutation. Defects in one such gene, *MUTYH*, are known to elevate the incidence of colorectal cancer in a recessive Mendelian manner. Recent evidence has also linked *MUTYH* to a mutator phenotype affecting normal somatic cells as well as the female germline. Here, we use whole-genome sequencing to measure germline de novo mutation rates in a large extended family containing both mothers and fathers who are affected by pathogenic *MUTYH* variation. By developing novel methodology that uses siblings as "surrogate parents" to identify de novo mutations, we were able to include mutation data from several children whose parents were unavailable for sequencing. In the children of mothers affected by the pathogenic *MUTYH* genotype p.Y179C/V234M, we identify an elevation of the C>A mutation rate that is weaker than mutator effects previously reported to be caused by other pathogenic *MUTYH* genotypes, suggesting that mutation rates in normal tissues may be useful for classifying cancer-associated variation along a continuum of severity. Surprisingly, we detect no significant elevation of the C>A mutation rate in children born to a father with the same *MUTYH* genotype, and we similarly find that the mutator effect of the mouse homolog *Mutyh* appears to be localized to embryonic development, not the spermatocytes. Our results suggest that maternal *MUTYH* variants can cause germline mutations by attenuating the repair of oxidative DNA damage in the early embryo.

**Keywords:** germline mutagenesis; de novo mutation calling; mutational signature; *MUTYH*-associated polyposis; mutator allele; oxidative damage; pedigree

## Introduction

Many DNA repair deficiencies are linked with increased risk for cancer syndromes (Fearon 1997; Goode *et al.* 2002; Matullo *et al.* 2006; Randall *et al.* 2024). Pathogenic mutations leading to the loss of function in specific DNA repair pathways accelerate the accumulation of oncogenic variants. While each DNA repair defect often tends to cause cancers mainly in specific tissues, other tissues may also accumulate a higher mutation load than normal (Dunlop *et al.* 1997; Aarnio *et al.* 1999; Scarbrough *et al.* 2016). It is not well understood why accelerated mutagenesis only seems to lead to cancer in certain tissues, or whether somatic mutations that do not cause cancer might have other health impacts (Elledge and Amon 2002; Chao and Lipkin 2006; Blokzijl *et al.* 2016).

Some recent studies (Kaplanis *et al.* 2022; Sherwood *et al.* 2023; Andrianova *et al.* 2023; Stendahl *et al.* 2023) have paid particular attention to the impact of DNA repair deficiencies on the germline because even modestly elevated germline mutation rates can impact congenital disease risk and the rate of evolution. Moreover, since germline mutations can be studied through relatively straightforward comparisons between relatives (Wei *et al.* 2015; Bergeron *et al.* 2022) and do not require the specialized technologies that are needed to detect low-frequency somatic variants (Kennedy *et al.* 2014; Ellis *et al.* 2021), germline mutator phenotypes have the potential to lead to discovery of new DNA repair defects that may be candidate drivers of novel cancer syndromes. For example, inherited variation was recently used to discover that a variant in the murine base excision repair (BER) DNA-glycosylase *Mutyh* gene acts as a germline mutator allele in inbred mouse strains (Sasani *et al.* 2022, 2024). Since impaired functioning of the human MUTYH protein is known to cause a colorectal cancer syndrome known as *MUTYH*-associated polyposis (MAP) (Smith, West, *et al.* 2013), this mutator allele is a promising candidate for exploring joint effects of DNA repair genes on the mammalian soma and germline.

The *MUTYH* gene plays a key role in BER, a pathway that repairs damage caused by reactive oxygen species (ROS), which are byproducts of aerobic metabolism (Banda *et al.* 2017). ROS can react with guanine to create the lesion 8-oxoguanine (8-OG), which has a propensity to mispair with adenine, resulting in G:C>T:A transversion mutations, often abbreviated as C>A mutations (David *et al.* 2007). BER DNA glycosylases have developed a specific

mechanism to repair this mutagenic damage: OGG1 removes 8-OG from the compromised strand (Hayashi *et al.* 2002) while MUTYH excises the erroneously incorporated adenines opposite 8-OG (Krokan and Bjørås 2013; Woods *et al.* 2016). Due to MUTYH's role in this repair pathway, defects in this enzyme can cause excess accumulation of C>A mutations in tissues that are experiencing ROS damage (Pilati *et al.* 2017).

MAP follows a recessive inheritance pattern, affecting individuals who have inherited 2 subfunctional copies of the *MUTYH* gene (Morak *et al.* 2014), causing intestinal adenomatous polyposis as well as an elevated risk for early-onset colorectal and duodenal malignancies (Al-Tassan *et al.* 2002; Nielsen *et al.* 2011). The genotypes that cause MAP are often referred to as "biallelic": biallelic *MUTYH* genotypes are either homozygous for a single pathogenic variant or are compound heterozygotes, with each copy of *MUTYH* affected by a different pathogenic mutation. In contrast, individuals who have "monoallelic" genotypes (meaning heterozygous for a single pathogenic *MUTYH* variant) do not generally develop intestinal polyposis and have at most a modestly elevated cancer risk (Barreiro *et al.* 2022). Notably, *MUTYH* is an example of a gene that plays a crucial role in genomic stability across all tissues affected by ROS damage, but mainly appears to modulate cancer risk in the colorectal epithelium (Nieuwenhuis *et al.* 2012; Hutchcraft *et al.* 2021). Despite the tissue specificity of MAP's cancer risk phenotype, recent evidence indicates that this condition also causes elevated somatic mutation rates in a wider variety of human cell types, including blood (Robinson *et al.* 2022), which might be why some studies have found *MUTYH* variants to be associated with increased risk of extracolonic cancers (Zhang *et al.* 2006; Beiner *et al.* 2009; Vogt *et al.* 2009; Win *et al.* 2016; Villy *et al.* 2022). These findings led us to hypothesize that *MUTYH*'s C>A mutator effect might extend to germline cells.

To test whether pathogenic *MUTYH* genotypes might cause a human germline mutator phenotype, we sequenced 15 genomes from a large extended family containing multiple individuals affected by MAP. We used these genomes to test whether parent–child trios including a parent with MAP show evidence of elevated C>A mutation rates compared to trios where the parental *MUTYH* genotypes are normal or monoallelic. Recently, another group published de novo mutation (DNM) data from 2 trios whose mothers were affected by MAP (Sherwood *et al.* 2023), obtaining evidence for a C>A mutator effect. We further investigated this effect in our larger data set, which includes children of both mothers and fathers affected by the same pathogenic *MUTYH* genotype. We then further contextualized our results through a comparison to a null model of mutation rate as function of parental age that was previously constructed from thousands of control trios (Jónsson *et al.* 2017). In this way, we were able to characterize how pathogenic *MUTYH* variants affect the human germline, using an analysis framework that is broadly appropriate for investigating the effects of other cancer syndromes on germline mutagenesis and human evolution.

## Materials and methods
### Recruitment and consenting of study subjects for biospecimen collection
The design of this study received prior approval from the University of Washington Institutional Review Board. After study participants gave written informed consent, they were each mailed an OGR-500 Oragene saliva collection kit. DNA was extracted from the Oragene kits at the Fred Hutchinson Cancer Center specimen processing core facility using the recommended

standard protocol and then sequenced to 50X coverage at the Northwest Genome Center.

### SNP calling and trio-based DNM calling
Variant detection and genotyping were performed using the GATK HaplotypeCaller (4.2.0.0) (Van der Auwera 2020). Variants were initially flagged using the filtration walker (GATK) to mark sites that were of lower quality [e.g. low-quality scores (Q50), allelic imbalance (ABHet 0.75), long homopolymer runs (HRun > 4), and/or low quality by depth (QD < 5)]. Data QC included an assessment of (1) mean coverage; (2) fraction of genome covered greater than 10X; (3) duplicate rate; (4) mean insert size; (5) contamination ratio; (6) mean Q20 base coverage; (7) transition/transversion ratio (Ti/Tv); (8) fingerprint concordance > 99%; (9) sample homozygosity and heterozygosity; and (10) sample contamination validation. Genome completion was defined as having >95% of the target read at >10X coverage and >90% of the target at >20X coverage.

Putative DNMs in parent–offspring and surrogate–offspring trios were identified using the GATK (v4.2.6.1) PossibleDeNovo tool, which uses the genotype information from individuals in family trios to identify possible DNMs and the sample(s) in which they occur.

### Using surrogate parents for DNM calling
In children without 2 sequenced parents, we called DNMs using parental haplotypes shared with "surrogate parent" siblings. To identify parental haplotypes shared between relatives, we began by phasing the full 15-genome data set using Beagle (Browning *et al.* 2021). In order to improve phasing quality, we phased these genomes together with a panel of 3,202 genomes from the high-coverage 1000 Genomes Project (Byrska-Bishop *et al.* 2022). Since rare variants are generally uninformative for identity-by-descent (IBD) segments and are prone to sequencing error and phasing error, we filtered for common variants that are found at minor allele frequency > 10% in a subset of 2,504 genomes and used them as the input to the program hap-IBD to infer shared tracts of IBD (Zhou *et al.* 2020). Additionally, the following hap-IBD parameter settings were used: min-seed=1.0, max-gap=1000, min-extend =0.2, min-output=2, min-markers=100. In this way, we were able to identify IBD segments that were shared between siblings but not present in any sequenced parent and then use these IBD segments as surrogates for missing paternal and maternal genome sequences. We noted that putative DNMs often occurred near the ends of our inferred surrogate parent tracts, and we hypothesized that these might be artifacts caused by inaccuracies in the boundaries of shared IBD tracts. To eliminate these artifacts, we implemented a density-based filter (see *Filtering* in the Supplementary Methods and Fig. 1b).

In some probands, we called DNMs using a mother plus sibling surrogate fathers, restricting to regions where the proband and the sibling share paternal DNA. In other probands, we called DNMs using 2 surrogate parent siblings, restricting to regions where the proband shares each of their haplotypes IBD with one or more siblings (for specific details on the specific surrogate parent configurations that were used, see *Using surrogate parents for DNM calling* in the Supplementary Methods). Within these accessible regions, DNMs were called using GATK PossibleDeNovo, substitution surrogate parents for one or both parents. All preliminary DNM calls were then filtered as described in *Filtering* and *IGV inspection* in the Supplementary Methods. During the IGV inspection step, we eliminated any putative DNM shared between 2 or more individuals, except when those individuals have a parent–child relationship, assuming that most such shared

variants were in fact inherited from unsequenced parents in regions erroneously identified as inherited IBD. To eliminate additional false positive calls from the data, we manually removed any set of mutations that were clustered (over 6 mutations within 50 bp of one another) near a putative DNM with an rsID annotation, as we inferred these sites were more likely to have been inherited from a missing parent rather than a multinucleotide germline mutation event.

## Generation and analysis of simulated trio data for surrogate method benchmarking

To benchmark the performance of the surrogate method, we simulated a realistic pattern of mutation accumulation in a 5-sibling family and then simulated sequencing reads consistent with the family's genotypes. In brief, we generated the family's inherited germline variation using the 1000 Genomes high-coverage phased data set (Byrska-Bishop *et al.* 2022) and then generated DNMs based on the data from Jónsson *et al.* (2017), both mapping to the hg38 assembly. Only autosomes were simulated.

More specifically, we sampled one "parent" at random from the 1000 Genomes CEU population and sampled the other at random from the GBR population (the specific sampled individuals were NA11893 and HG00132). This sampling was designed to match the European ancestry of our pedigree but avoid sampling individuals who were too closely related. We randomly generated 5 "children" of these parents by sampling recombination breakpoints from a chromosome map and adding the DNMs observed in a child sequenced in a previous study by Jónsson *et al.* (2017). Finally, we generated a short-read BAM file consistent with each simulated genome using DWGSIM (0.1.15, www.github.com/nh13/DWGSIM), a tool that simulates sequencing reads from a reference genome and can incorporate custom germline variants. For additional details, see *Generation and analysis of simulated trio data for surrogate method benchmarking* in the Supplementary Methods.

## Calculation of each family's expected mutation burden in the absence of a genetic mutator effect

Jónsson *et al.* (2017) carried out whole-genome sequencing of Icelandic families and identified parental age impacts on the number and spectra of inherited DNMs. We used the Poisson regressions carried out in this study (listed in Table S9 of Jónsson *et al.* 2017) to predict expected DNM burdens and spectra for each of the families in our study, based on parental ages.

For each individual in our study, we plugged their parents' paternal and maternal ages at the time of their birth into the following equations to get the expected count of each mutation type $c$ (C>A, C>T, C>G, A>G, A>C, A>T):

$$y_{c,\text{mat}}(a_{\text{mat}}) = m_{c,\text{mat}} \cdot a_{\text{mat}} + b_{c,\text{mat}}$$

$$y_{c,\text{pat}}(a_{\text{pat}}) = m_{c,\text{pat}} \cdot a_{\text{pat}} + b_{c,\text{pat}}$$

In these equations, $a_{\text{mat}}$ and $a_{\text{pat}}$ are the maternal and paternal ages at the time of a child's birth, respectively, $m_{c,\text{mat}}$ and $m_{c,\text{pat}}$ are the numbers of mutations of type $c$ accumulated each year in the maternal and paternal germlines [linear regression slopes from Jónsson *et al.*'s (2017) Table S9], and $b_{c,\text{mat}}$ and $b_{c,\text{pat}}$ are the numbers of mutations of type $c$ that would theoretically be present in the maternal and paternal germlines at age 0 (mutation type-specific maternal and paternal linear regression $y$-intercepts). The resulting expected DNM counts inherited from the mother and father add up to the expected burden of each

type of DNMs in their child. For additional details on correcting the accessible genome size and application of the model to additional data sets, see *Calculation of each family's expected mutation burden in the absence of a genetic mutator effect* in the Supplementary Methods. For details on statistical analyses of deviation from this null model, see the additional Supplementary Methods sections *Comparing our observed mutation counts to the null parental age model of Jónsson et al. (2017)* and *Estimating the minimum mutator effect sizes that we have power to detect.*

## Cellular assay of MUTYH glycosylase function

Human HEK293 *MUTYH* KO cell lines were transduced with lentivirus containing *MUTYH* cDNAs, either WT or variant, each cloned into pCW57.1 (Addgene #41393; gift from Dr. David Root). Transduced cells were selected, and stable MUTYH expression was induced as previously described (Jia *et al.* 2021). To measure *MUTYH* variant function, cells expressing each variant were then co-transfected with a GFP reporter containing an 8oxoG:A mispair (Raetz *et al.* 2012; Nagel *et al.* 2014) and an mCherry-expressing plasmid as a transfection control. After a ~72 h incubation with the reporter, cells were analyzed via FACS with a BioRad Ze5. A function score was calculated as the fraction of repair positive (mCherry+, GFP+) cells out of all transfected cells (mCherry+), divided by the same quantity for cells transduced with WT *MUTYH*, and scaled by a $\log_2$ transform, such that a score of 0 indicates WT-like repair function, and negative scores indicate deficient function.

## Analysis of parental age dependence of the BXD mouse mutation rate

We downloaded data on the mutation rate per generation, B vs D haplotype status at the *Mutyh* locus, and the number of years of inbreeding, and the number of generations of inbreeding for each of the BXD mouse strains from the Github page associated with Sasani *et al.* (2022) (https://github.com/tomsasani/bxd_mutator_manuscript). We calculated the average generation time of each strain to be its number of years of inbreeding divided by its number of generations of inbreeding. We then computed each strain's C>A mutation rate per site per generation (number of C>A mutations divided by generations of inbreeding divided by the accessible genome size) and fit an ordinary least square multilinear regression to explain this variable as a function of generation time minus the minimum observed generation time of 0.2 years as well as the categorical *Mutyh* status variable.

## Results

### Whole-genome sequencing of a family affected by a MUTYH genotype that shows reduced DNA repair efficiency in vitro

We obtained saliva samples from 3 full siblings (labeled P1, P2, and P3 in Fig. 1) who share a biallelic *MUTYH* genotype known as p.Y179C/V234M. One gene copy has a tyrosine-to-cysteine substitution at amino acid position 179, and the other has a valine-to-methionine substitution at position 234. Both amino acid positions are indexed in the coordinates of *MUTYH* transcript NM_001128425; substitutions p.Y179C and p.V234M have DNA coordinates c.536A>G and c.700G>A, respectively. We also sampled saliva from a fourth sibling (P4 in Fig. 1) who is a monoallelic carrier of p.V234M. These siblings inherited p.Y179C from their father and inherited p.V234M from their mother. Two of the 3 biallelic sibling individuals were previously diagnosed with colon cancer, and the third has a history of colon polyps (all 3 meet the diagnostic criterion for MAP by virtue of their *MUTYH* genotype). The siblings' extended

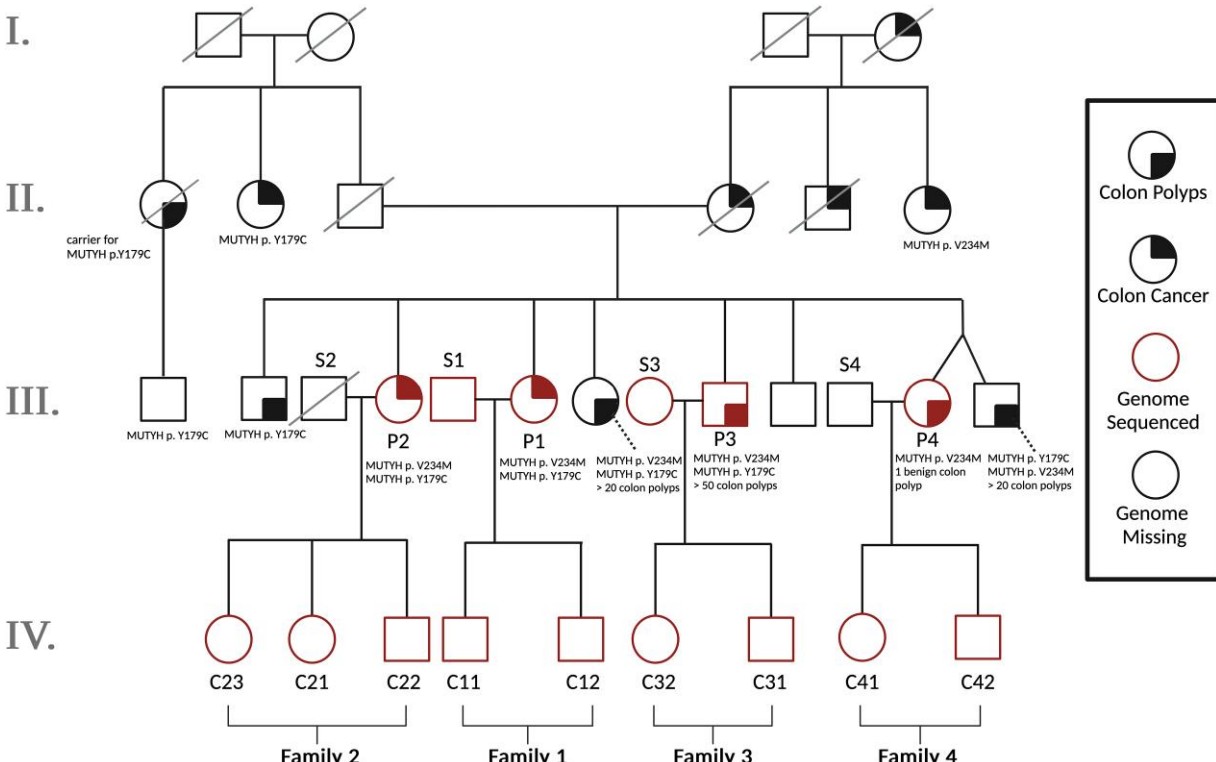

**Fig. 1.** Pedigree of all sequenced individuals plus unsampled relatives. To measure the effects of biallelic *MUTYH* mutations (present in Generation III) on germline mutagenesis, we sequenced all offspring from generation IV as well as all parents from Generation III from whom DNA samples were available. Individuals have been given labels according to whether they are part of nuclear families 1–4, whether they are a *MUTYH* variant carrier parent (P), a spouse or partner of that parent (S), or a child (C). The sequenced individuals make up 4 nuclear families whose children are all first cousins: Families 1 and 2 include mothers with the biallelic genotype p.Y179C/V234M, while Family 3 includes a father with the same p.Y179C/V234M genotype. Family 4 includes a mother who is monoallelic for p.V234M. Parents of the Generation III individuals are each carriers of one of these monoallelic mutations. Solid black quadrants indicate which individuals have been diagnosed with colon polyps (bottom right) or colon cancer (top right). *MUTYH* mutations and number of identified colon polyps are listed below individuals studied in this pedigree. Square, male; circle, female; red, genome sequenced; black, genome missing from sequenced pedigree trio.

family is affected by a notably elevated level of colorectal cancer, including unsampled relatives who are also pictured in Fig. 1. All sampled individuals tested negative for Lynch syndrome variants and other variants known to cause heritable colorectal cancer.

While ClinVar classifies p.Y179C as pathogenic with evidence from many previous studies (Al-Tassan *et al.* 2002; Nielsen *et al.* 2005, 2009; Vogt *et al.* 2009), some laboratories consider p.V234M to be a variant of uncertain significance with mixed functional evidence (Fleischmann *et al.* 2004; Peterlongo *et al.* 2006; Komine *et al.* 2015; Yurgelun *et al.* 2015). To obtain more information about the pathogenicity of the genotype p.Y179C/V234M, we conducted functional assays in which mutant *MUTYH* expression is restored in human HEK293 *MUTYH* KO cells. Our approach uses a reporter construct engineered to contain an 8-oxoG:A lesion, such that proper repair corrects a premature stop codon in GFP and restores its expression. Notably, the p.Y179C allele exhibited severe loss of repair function, whereas the p.V234M variant displayed a partial loss of function with repair activity well below that of wild-type MUTYH (Supplementary Fig. 2). The deleterious effects observed for these 2 variants within the HEK293 cell context indicate that they likely have pathogenic effects and may result in elevated mutation accumulation across tissues in vivo.

We also used the same functional assay to measure the effects of c.461GT>AA p.R182Q (corresponding to the DNA substitution c.461 GT>AA on transcript NM_001128425), the human analog

of the mutation found in an outlier mouse strain known as BXD68 that displayed a *Mutyh* hypermutator germline phenotype (Sasani *et al.* 2022). p.R182Q appears to be a total loss of function variant with a phenotype similar to that of p.Y179C.

We note that the previous study of *MUTYH*'s germline mutator activity by Sherwood *et al.* (2023) analyzed families with a different compound heterozygous genotype known as p.Y179C/G368D. p.G368D (corresponding to the DNA substitution c. 1187 G>A on transcript NM_001128425) may be less deleterious than p.Y179C given its association with an older age at MAP diagnosis (Guarinos *et al.* 2014) and its less severe somatic mutator phenotype (Robinson *et al.* 2022). Note that Robinson *et al.* refer to the variant p.G368D as p.G396D in the coordinates of a different reference *MUTYH* transcript.

To assess the impact of the *MUTYH* p.Y179C/V234M genotype on the germline mutation rate and spectrum, we performed 50X coverage whole-genome sequencing on P1–P4 as well as 9 of their adult children and 2 of their spouses (Fig. 1). Individuals have been given labels according to which nuclear family they are a member of (1–4), and whether they are a *MUTYH* variant carrier parent (P), a spouse or partner of that parent (S), or a child of a *MUTYH* variant carrier (C). Colloquially, we refer to all parents as mothers and fathers if they conceived their children via oocytes and spermatocytes, respectively, although we recognize that these labels may not match parents' individual gender identities.

## Using siblings as "surrogate parents" for DNM calling in incomplete nuclear families

DNMs are typically called by identifying sites that violate the principles of Mendelian inheritance. These are sites at which a child's genome contains a variant not observed in the genome of either of their parents. DNM calling normally requires the genomes of both parents to be sequenced, and Families 1 and 3 are the only nuclear families in our data set that meet this requirement (Fig. 1). Families 2 and 4 were not suitable for standard mutation calling due to unavailability of the paternal genomes (one father is deceased and the other declined to participate). To maximize the power of this study, we developed a novel method that facilitated DNM calling in these families. Although our new method has slightly lower accuracy and precision than standard DNM calling, particularly in Family 4 that consists of just a mother and 2 children, it enables DNM analysis in previously inaccessible families within our study and potentially beyond.

Since each child in Generations III and IV has at least one full sibling represented in our data set, we were able to leverage the sharing of parental haplotypes among siblings to devise a "surrogate parent" method for estimating DNM rates and spectra. Instead of comparing each child's genome to their mother and father to identify variants that must have arisen de novo, we compared each child from Families 2 and 4 to their mother plus one or more siblings who inherited some of the same paternal DNA. If the child's genome contains a variant that is absent from both the mother and the sibling surrogate father, this implies that the unique variant arose de novo in the child (Fig. 2a and b; pipeline described in Supplementary Fig. 1). Jónsson *et al.* (2018) previously used a similar procedure to identify mutations that arose early in parental embryonic development, which often have parental read support and are thus not detectable by standard parent–child trio mutation calling methodology.

As in standard DNM calling pipelines (Bergeron *et al.* 2022), we only call mutations at sites where both the real and surrogate parents are homozygous for the reference allele. This allows us to polarize mutation calls with confidence: if we are calling DNMs in sibling 1 using sibling 2 as a surrogate parent, all DNM calls will occur at sites where sibling 1 is heterozygous and sibling 2 is homozygous, which are not variants that could also be explained by DNMs in sibling 2.

Our use of the term "surrogate parent" is loosely based on the established use of relatives as surrogate parents for haplotype phasing (Kong *et al.* 2008). We also drew inspiration from several recent studies that successfully estimated mutation rates using tracts inherited IBD between relatively distant relatives (Narasimhan *et al.* 2017; Tian *et al.* 2019, 2022). In contrast to these earlier studies, which estimated population-averaged mutation rates using mutations that likely occurred many generations ago, our method is designed to estimate an individual parent–child trio's mutation rate using mutations that arose within a single generation. This method enabled us to estimate germline mutation rates within 5 nuclear subfamilies of the pedigree and study how *MUTYH* genotype affected the germline mutation rate and spectrum.

To identify regions accessible for DNM calling using a mother and a surrogate father, we first used the software hap-IBD (Zhou *et al.* 2020) to identify long haplotypes shared identical by descent between each pair of siblings. We then filtered these regions to exclude maternally inherited haplotypes. In addition, we implemented a "SNP density filter" to exclude what appear to be false positive IBD calls: regions where the density of pairwise differences between the siblings is too high to be consistent with true sharing of paternal haplotypes (see *Filtering* in the Supplementary Methods). Within the remaining regions of paternal haplotype sharing, we called DNMs using a standard GATK-based pipeline, using the surrogate father as the father. In this setup, gene conversion between the 2 paternal haplotypes has the potential to create false positive DNMs, as previously observed by Narasimhan *et al.* (2017). To minimize these errors, we filtered out putative DNMs that were present in the 1000 Genomes data or in 2 or more members of our pedigree (this should eliminate gene conversion errors at any loci where not all siblings inherited the same paternal haplotype). We note that the familial mutation sharing filter will cause us to miss the 1–2% of DNMs that are shared between siblings due to germline mosaicism (Jónsson *et al.* 2018). The 1000 Genomes filter may also cause us to exclude some true DNMs, particularly at CpG sites or other mutation hotspots, but it should effectively filter out many false positive DNMs that were actually inherited from a missing parent, except when those inherited variants are very rare in the population as a whole.

In a large family with a mother and $n + 1$ siblings, the only regions inaccessible for surrogate father DNM calling will be regions where $n$ siblings inherited the same paternal haplotype and the remaining sibling inherited the other paternal haplotype. In this scenario, the sibling who inherited the unique paternal haplotype has no one to serve as a surrogate father, so that sibling's genome will be inaccessible for DNM calling while the $n$ other siblings' genomes will be accessible as long as $n$ is greater than 1. If we consider the paternal haplotype inherited at a specific locus in a specific sibling's genome, the probability that any given sibling inherited the same paternal haplotype is one-half. Therefore, the probability that all $n$ other siblings inherited the other paternal haplotype is $2^{-n}$. If any one other sibling did inherit the same paternal haplotype, the first sibling's genome will be accessible for DNM calling at the locus we are considering. This implies that in our hypothetical family with one parent and $n + 1$ children, a fraction $1 - 2^{-n}$ of all DNMs should be callable. For example, in Family 2, which consists of 3 children and their mother, three-fourths of all DNMs should be callable.

Within each region that is accessible for DNM calling, most DNMs occurring on the proband's maternal and paternal haplotypes should be identifiable, with the exception of DNMs that are shared with the sibling surrogate parent. Neglecting these sib-shared mutations, the mutation rate can be estimated by dividing the mutation count by 2 times the length of the accessible genomic region spanned by paternal IBD tracts. Moreover, read-backed phasing tools that are designed for use in parent–child trios can be applied with the surrogate father substituted for the father (Belyeu *et al.* 2021). Read-backed phasing will deduce that a mutation arose on a paternally inherited chromosome if it can be phased to a haplotype shared between the siblings that is not shared with their mother. Similarly, we can deduce that a mutation occurred on a maternally inherited chromosome if it can be phased to a haplotype that is shared with the maternal genome.

Since full siblings share DNA inherited from both of their parents, they can serve as surrogate parents for DNM calling even when the mother and father are both deceased, as is the case for the Generation III siblings P1, P2, P3, and P4 (illustrated in Fig. 3a; for more details, see *Using surrogate parents for DNM calling* in *Materials and methods* and Supplementary Methods). Since maternal and paternal chromosomes are passed down independently of one another, the fraction of DNMs accessible for calling in a family with $n + 1$ children and no parents is expected to be

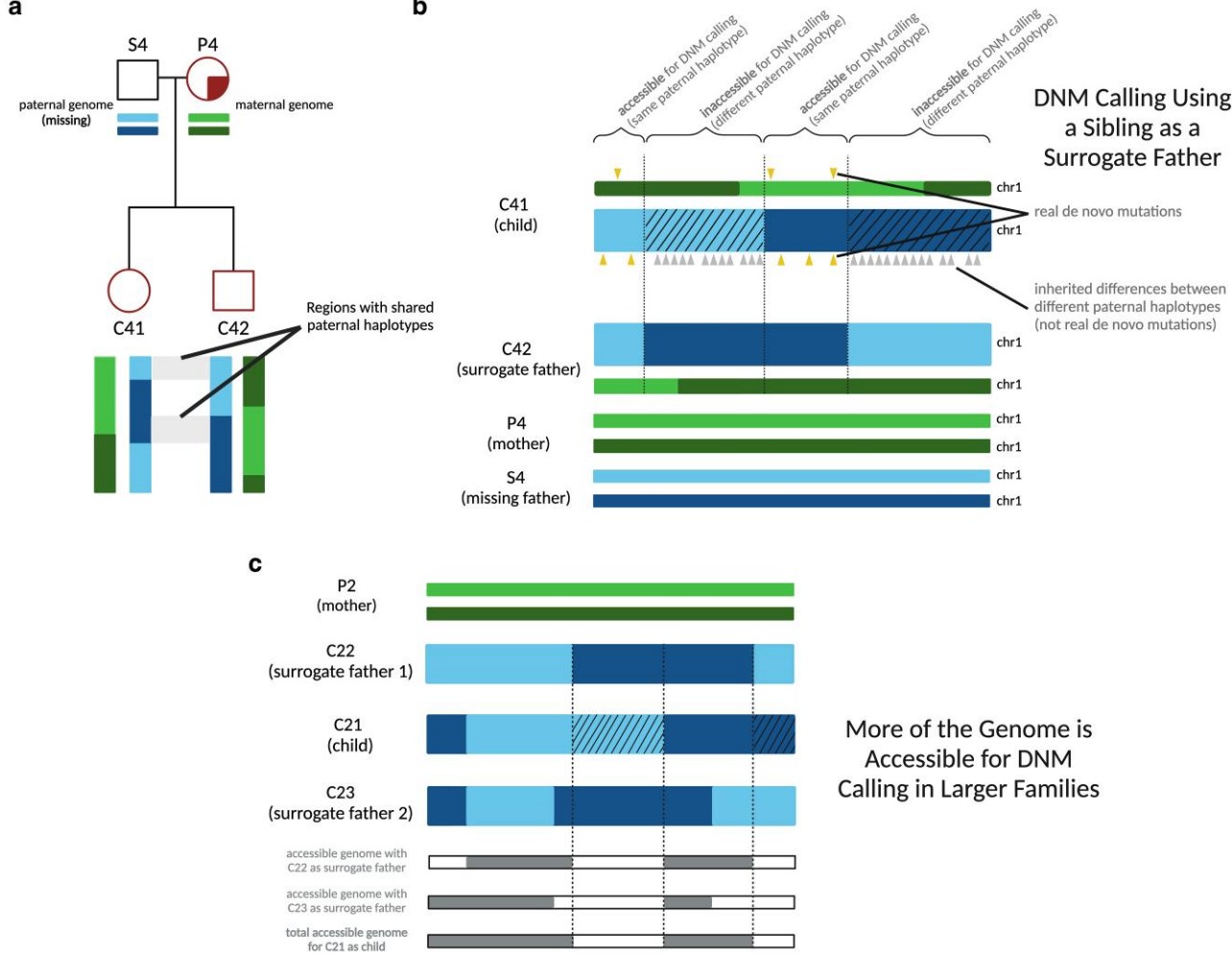

**Fig. 2.** Calling DNMs using 1 parent and 1 surrogate parent. a) An illustration of the portions of an autosome with paternal haplotypes shared between 2 siblings. In an example chromosome from Family 4, DNMs can be called in regions where C41 and C42 share a paternal haplotype sequence with one another. b) An illustration of DNM calling using a sibling as surrogate father. In regions where the siblings inherited the same paternal haplotype, Mendelian violations (DNM calls, yellow triangles) are spaced far apart, but in regions where the siblings inherited different paternal haplotypes, Mendelian violations (gray triangles) are clustered close together, mostly stemming from polymorphic differences between the different paternal haplotypes inherited by the respective siblings. Hashed chromosome regions represent inaccessible regions of the genome, where DNMs cannot be called using the surrogate approach. c) An example of the surrogate method applied to Family 2, a 3-child family where 2 different surrogate fathers can be used to call DNMs in each child. A set of partially overlapping candidate DNMs is generated from each sibling comparison, increasing the amount of accessible genome where mutations can be identified with more siblings used in this approach and allowing additional validation of calls in regions where accessible regions overlap.

$(1 - 2^{-n})^2$, slightly smaller than the fraction $1 - 2^{-n}$ that is callable when paternal or maternal DNA is available. To call DNMs in P1, P2, P3, and P4 using surrogate parents alone, we first used hap-IBD to identify tracts shared identical by descent between pairs of siblings. For each sibling trio ($P_i$, $P_j$, and $P_k$), we then identified the set of regions where DNMs are callable in $P_k$ using $P_i$ and $P_j$ as surrogate parents: this is the set of regions where $P_k$ shares one haplotype IBD with $P_i$ and shares the other haplotype IBD with $P_j$ (Fig. 3b). As long as 2 surrogate parents share distinct overlapping IBD tracts with the proband, we are able to randomly designate one as the surrogate mother and the other as the surrogate father and proceed with DNM calling without determining which haplotypes are actually maternally and paternally inherited (though with two surrogate parents we cannot use read-backed phasing to deduce whether mutations occurred on maternal vs paternal haplotypes). We also identified regions of the genome where one sibling was able to serve simultaneously as surrogate

mother and father to another sibling; this is possible wherever the two siblings inherited the same haplotype from each of their parents, which occurs across about 25% of the genome in full siblings (Fig. 3c).

To assess the quality of our surrogate-based DNM calls, we performed benchmarking using a family that we simulated as a composite of whole-genome sequences from the 1000 Genomes Project and mutation and recombination events from trios previously sequenced by Jónsson *et al.* (2017) (see *Generation and analysis of simulated trio data for surrogate method benchmarking* in *Materials and methods* and Supplementary Methods). We simulated a family with 5 total children, randomly selected one child as the "proband," and called DNMs in this child using different subsets of the available real and surrogate parents. In both real and simulated families, DNM calling accessibility was similar to our theoretical prediction (Fig. 4a). As illustrated in Fig. 4b, different sets of surrogate parents provided coverage of

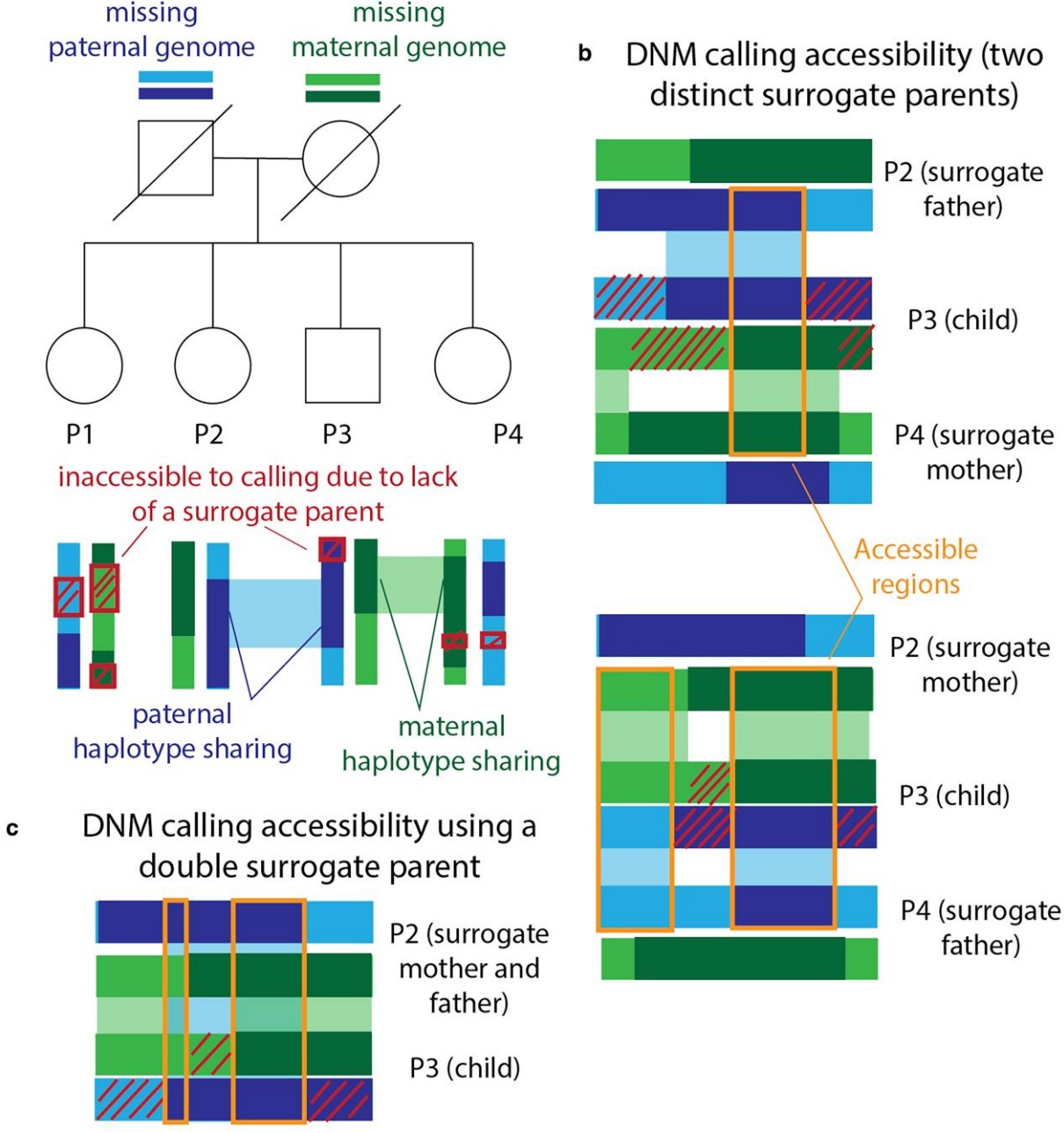

**Fig. 3.** Calling DNMs using only surrogate parents. a) Generation III of our pedigree contains 4 full siblings (P1, P2, P3, and P4) whose parents are deceased. We were able to call DNMs in these individuals using siblings as surrogate mothers and fathers. Example regions of paternal and maternal haplotype sharing are regions where P2 and P4 can act as surrogate father and mother to P3. Red crosshatches denote parental haplotype segments that were inherited by exactly 1 sibling—these regions are inaccessible to DNM calling due to lack of a surrogate parent. b) Illustration of maternal and paternal haplotype sharing in 2 example surrogate trios. DNMs are callable within orange regions due to haplotype sharing with both the surrogate mother and the surrogate father. c) Illustration of the regions that are accessible for calling in P3 using P2 as a double surrogate parent. This is possible when the surrogate and the proband share both maternal and paternal haplotypes.

complementary genomic regions, and overlap between these tracts permitted error correction of false positives that only appear when specific sets of surrogate parents are used. Overall inflation of the mutation rate by false positives is modest and appears to decrease with increasing family size (Fig. 4c–f). However, our simulation results suggest that caution is

warranted when interpreting the mutation rates and spectra in Family 4, which contains just 2 siblings as well as their mother. Only half of the genome is accessible for DNM calling in this family (Fig. 4a, Supplementary Fig. 3) and the ability to correct for paternal gene conversion will be limited, since none of the regions where DNMs are callable will contain both paternal haplotypes.

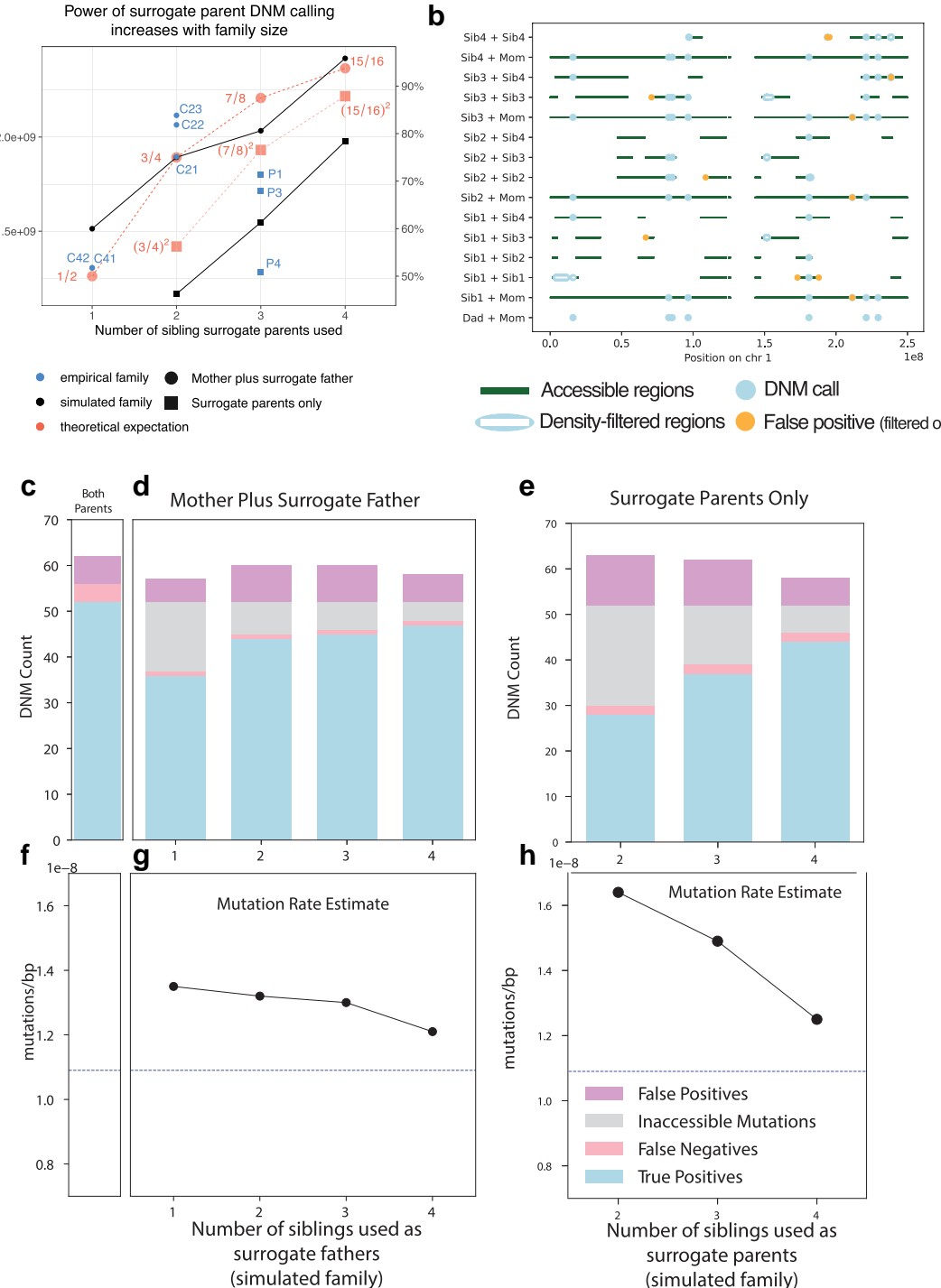

**Fig. 4.** Precision and recall of surrogate-based DNM calling. a) The number of base pairs accessible for DNM calling in each individual from our pedigree (blue scatter plot) depended on the number of real and surrogate parents available, broadly matching theoretical expectations (red dashed lines) as well as the proportion of accessible genome available for calling in our simulated pedigree (black solid lines) using similar configurations of real parents and siblings. Fractions in red indicate the proportion of the genome that is theoretically accessible. Families for which one real parent is available (circles) have a greater proportion of the genome accessible than those with no parents available (squares), but both improve when additional siblings are available. Theoretical expectations reflect the size of the genome accessible for SNP calling in the simulated family, which is close but not identical to the callable genome sizes in the empirical family. Empirical individual P2 was excluded due to somatic mutation contamination that lowered the accessible genome size (Supplementary Fig. 3; see discussion below). b) Genomic accessibility, DNM calls, and density-filtered regions in chromosome 1 of the simulated pedigree. Note that error correction is facilitated by overlap between regions accessible to calling using different sets of surrogate parents. c) True positive, false positive, and false negative DNM calls made in a simulated proband with both mother and father's genomes available. d) As in c), but with a mother plus different numbers of siblings to act as surrogate fathers. Inaccessible mutations are DNMs that occur in regions of the genome without a suitable surrogate father. e) True positive, false positive, and false negative DNM calls made in a simulated proband with no parental genomes available, using different numbers of surrogate mothers and fathers. f) Mutation rates estimated with both simulated parents' genomes available (dashed line). g) Mutation rates estimated using the mother's genome and surrogate fathers. h) Mutation rates estimated using surrogate mothers and fathers, with no parental genomes available.

## Children of pathogenic MUTYH carriers have normal germline mutation rates

Previous studies have found that *MUTYH* variants specifically increase the C>A mutation rate in a variety of species and cell types (Sasani *et al.* 2022; Robinson *et al.* 2022). Since C>A comprises only about 10% of human DNMs, even relatively large perturbations of the C>A mutation rate are not necessarily enough to push the overall germline mutation rate significantly above its normal range, as previously seen in mice as well as humans (Sasani *et al.* 2022; Sherwood *et al.* 2023). In keeping with this expectation, we found most trios in this study to have normal mutation rates ranging from $8.23 \times 10^{-9}$ to $2.14 \times 10^{-8}$ mutations per base pair per generation (Supplementary Table 1), comparable to the range between $7.9 \times 10^{-9}$ and $1.9 \times 10^{-8}$ expected in healthy individuals with parental ages between 15 and 50 based on a large previous study (Jónsson *et al.* 2017). However, we called a much higher frequency of mutations ($4.4 \times 10^{-8}$ mutations per site per generation) in the genome of P2, the biallelic mother of Family 2. Upon further examination, we found most of P2's mutations to have unusually low variant allele frequencies (VAFs), between 20 and 50%. All other individuals had mutation VAF distributions centered around 50%, as expected of germline mutations that arose on 1 of 2 parental haplotypes (Supplementary Fig. 4). P2's VAF skew suggests that most of their DNM calls are likely somatic mutations rather than germline mutations. Sherwood *et al.* (2023) previously noted a similar pattern in one of their biallelic *MUTYH* carriers who had undergone 5-fluorouracil chemotherapy for colorectal cancer, a treatment that can cause high-frequency mutations to emerge in the hematopoietic stem cell population. Due to this excess load of somatic mutations, which preclude estimation of an accurate germline mutation rate, we excluded P2 from further analysis and required a minimum VAF threshold of 30% for all mutations called in other individuals. Despite P2's high somatic mutation rate, we found that their mutation spectrum was dominated by normal background mutational signatures SBS1 and SBS5, with no sign of the *MUTYH*-associated signatures SBS18/SBS36 (Supplementary Fig. 5).

## Testing the children of biallelic MUTYH carriers for skewed mutation spectra and parent-of-origin bias

Although we did not expect the overall mutation rate to be significantly elevated in trios with biallelic *MUTYH* carrier parents, we hypothesized that these families might have elevated proportions of C>A mutations and/or a higher-than-expected proportion of C>A mutations inherited from the affected parent. To maximize our power to test for these effects, we calculated trio-specific expected C>A mutation counts and proportions using a model fit to patterns of DNMs in 1,548 Icelandic trios with no known mutator phenotypes (Jónsson *et al.* 2017). Although this control data set was generated separately from our study, we carried out similar filtering methodologies (Supplementary Fig. 1a), and all individuals in both studies are of European descent.

The Icelandic trio study by Jónsson *et al.* (2017) leveraged their data to predict the expected rate of each 1-mer mutation type per base pair per generation as a function of paternal and maternal age. Using this parental age model, we were able to calculate each trio's expected maternal and paternal 1-mer mutation burden as a function of the parents' ages at conception of the child (Supplementary Table 1) and their accessible genome sizes (Supplementary Fig. 6 and Table 1), following an approach recently used by Kaplanis, *et al.* (2022). For the most part, our empirical

counts agreed with these expected counts (Fig. 5a). In every child except for C42, the younger child with the abnormally high mutation rate in the family where we previously flagged DNM calling issues, the observed total mutation burden is within the upper 1-tailed Poisson 95% confidence interval expected under the parental age model (Supplementary Fig. 7).

When we categorized DNMs by 1-mer mutation type (Supplementary Table 2), we found that individual trio mutation spectra were largely consistent with the parental age model (Supplementary Fig. 9), but that C>A is the mutation type whose observed counts were most consistently inflated above expected counts, exceeding the expected count in all 12 individuals (Fig. 5b and c). Across the remaining five 1-mer mutation types, the proportion of trios exceeding the mutation count predicted by the parental age model ranged from 3/12 trios (A>C mutations) to 9/12 trios (C>T mutations) (Fig. 5b). Most of the elevated C>A counts fell within an upper 1-tailed 95% Poisson confidence interval of the expected count, but 3 children's C>A burdens significantly exceeded the parental age model expectation (Fig. 5c). These included C42 (one of our bioinformatic outliers), but also included C12 and C23, the children of 2 different biallelic mothers. The only non-C>A counts significantly exceeding the parental age model expectation were A>G mutations in C41 and C42 (a possible signal of inherited germline variant bleed-through due to the surrogate-calling method) and A>T mutations in C12 (Fig. 5c). For comparison, we analyzed the Sherwood *et al.* (2023) *MUTYH* trio data in the same framework and measured a significantly elevated DNM burden in 1 out of 2 children with a biallelic mother and well as much more significantly elevated DNM burdens in families with polymerase proofreading domain mutations (Supplementary Fig. 8).

We then added up sibling mutation counts to estimate the aggregate C>A enrichment within each nuclear family and found that the 2 families with biallelic mothers (Families 1 and 2) were enriched for C>A mutations by 1.81- and 1.46-fold above the expectation of the parental age model, respectively (Fig. 6a). In Family 1, the total C>A mutation burden significantly exceeded the 95% upper 1-tailed confidence interval of the parental age model, while Family 2 falls 1 mutation below this significance threshold (Fig. 6b). These C>A enrichments are comparable to the 1.57-fold-elevated C>A mutator phenotype recently identified in the mouse strain DBA/2J, but much less dramatic than the 6.04-fold enrichment phenotype identified in the mouse strain BXD68 caused by the homozygous loss of function R182Q-like mutation (Fig. 6a). In contrast, neither child with a biallelic father had a significantly elevated C>A mutation load, and their family (Family 3) was only enriched 1.34-fold for C>A mutations overall (Fig. 6a and b). C>A mutation load was only 1.14-fold elevated (also nonsignificant) in the 4 parents P1–P4, whose own parents were all monoallelic and thus not expected to have a germline mutator phenotype. In Family 4, the family with a biallelic mother and significant bioinformatic obstacles to accurate DNM calling, we observed a nonsignificant 1.65-fold C>A enrichment along with a significant 1.75-fold A>G enrichment (Fig. 6a and b). A 1.65-fold C>A enrichment fails to reach significance in Family 4 because the siblings C41 and C42 each have a smaller callable genome proportion than individuals from families with a father or third sibling available for genotype calling. As in Sherwood *et al.* (2023), we then further summed up the mutation counts for children with the same carrier parent type, which in the case of our pedigree means summing up Families 1 and 2 that both have a biallelic mother as the carrier parent. This biallelic mother group showed significant enrichment of C>A DNMs (Supplementary Fig. 10).

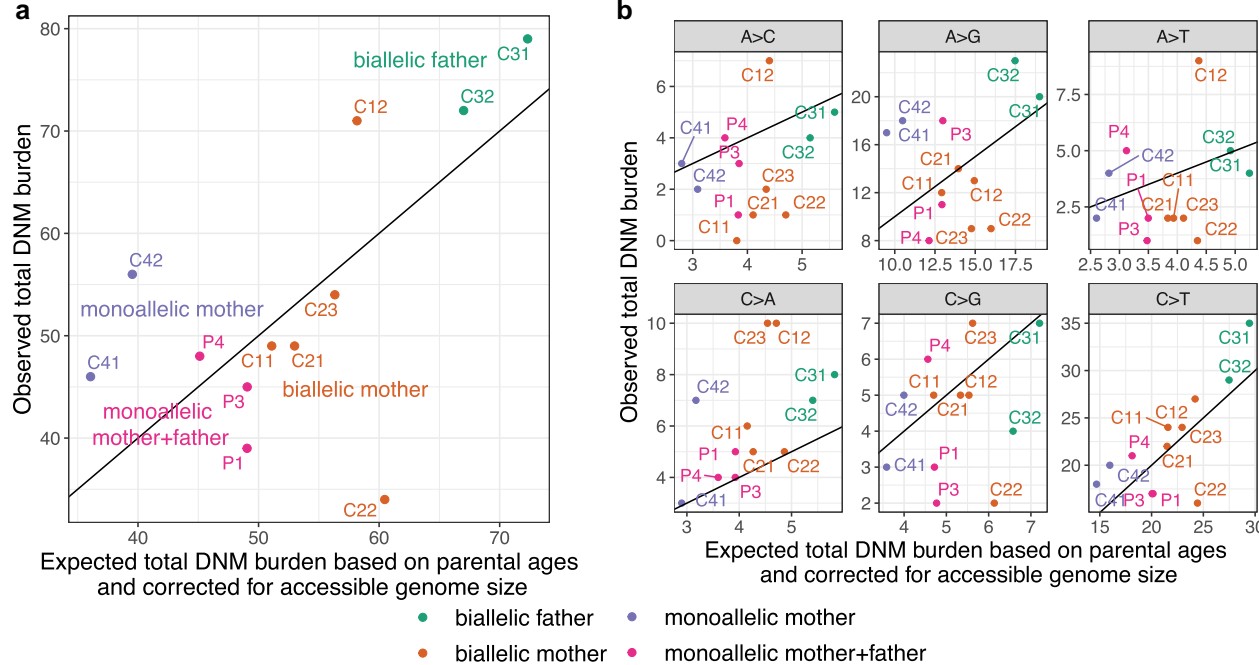

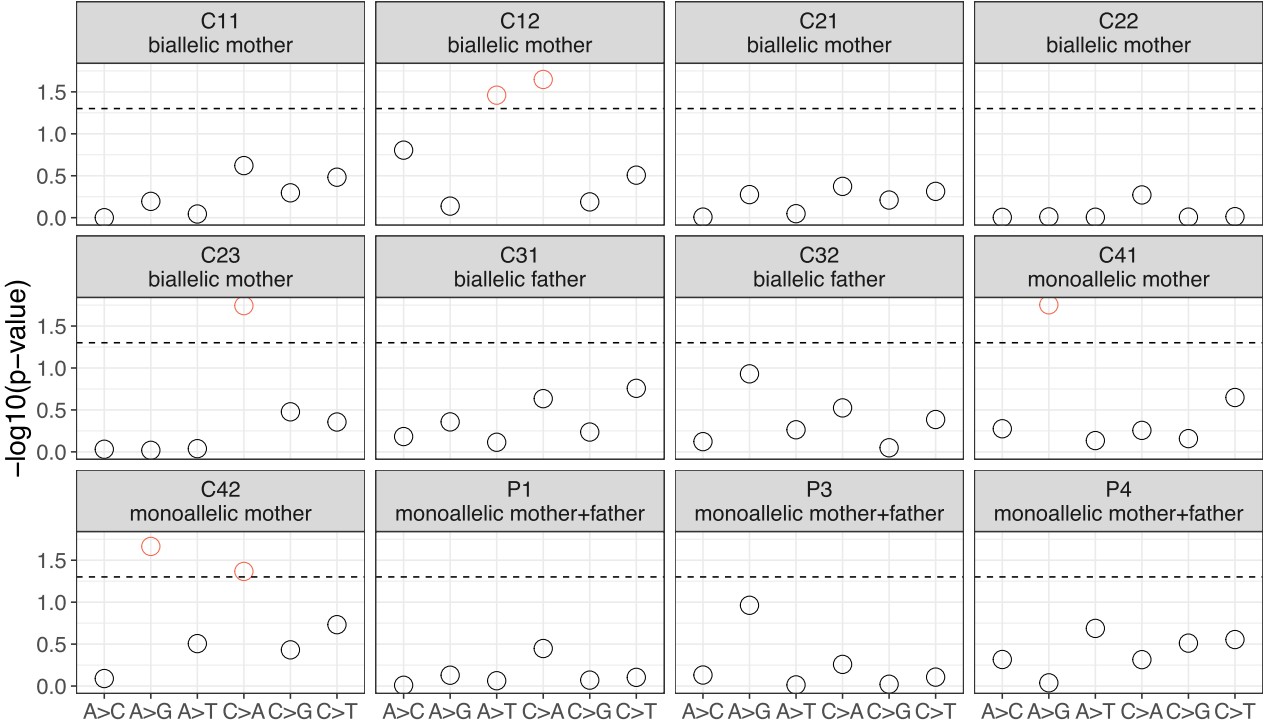

**Fig. 5.** Observed and expected mutation counts. a) Comparison of observed DNM counts per child and the corresponding expected DNM counts under the parental age model (Jónsson *et al.* 2017), corrected for accessible genome size (Supplementary Fig. 6). P2 was excluded as discussed above due to evidence for somatic mutation contamination. Points are colored by the *MUTYH* carrier status of the child's parent(s). Each child except for C42 has an overall mutation count that is compatible with the Jónsson parental age model (Supplementary Fig. 7). See Supplementary Fig. 8 for a comparison with the results of Sherwood *et al.* (2023). b) Observed and expected mutation counts, faceted by 1-mer mutation type. Note that C>A counts are above the $y = x$ line for nearly all individuals. c) The probability of observing a mutation count of each of the six 1-mer mutation types under the parental age model that is greater than or equal to what we observed for each member of the pedigree. Points above the dashed line (red circles) fall below the upper 1-tailed Poisson $P < 0.05$ significance threshold. C12 and C23, both children of biallelic mothers, show significant elevation of C > A DNM counts, as does C42 (child of a monoallelic mother).

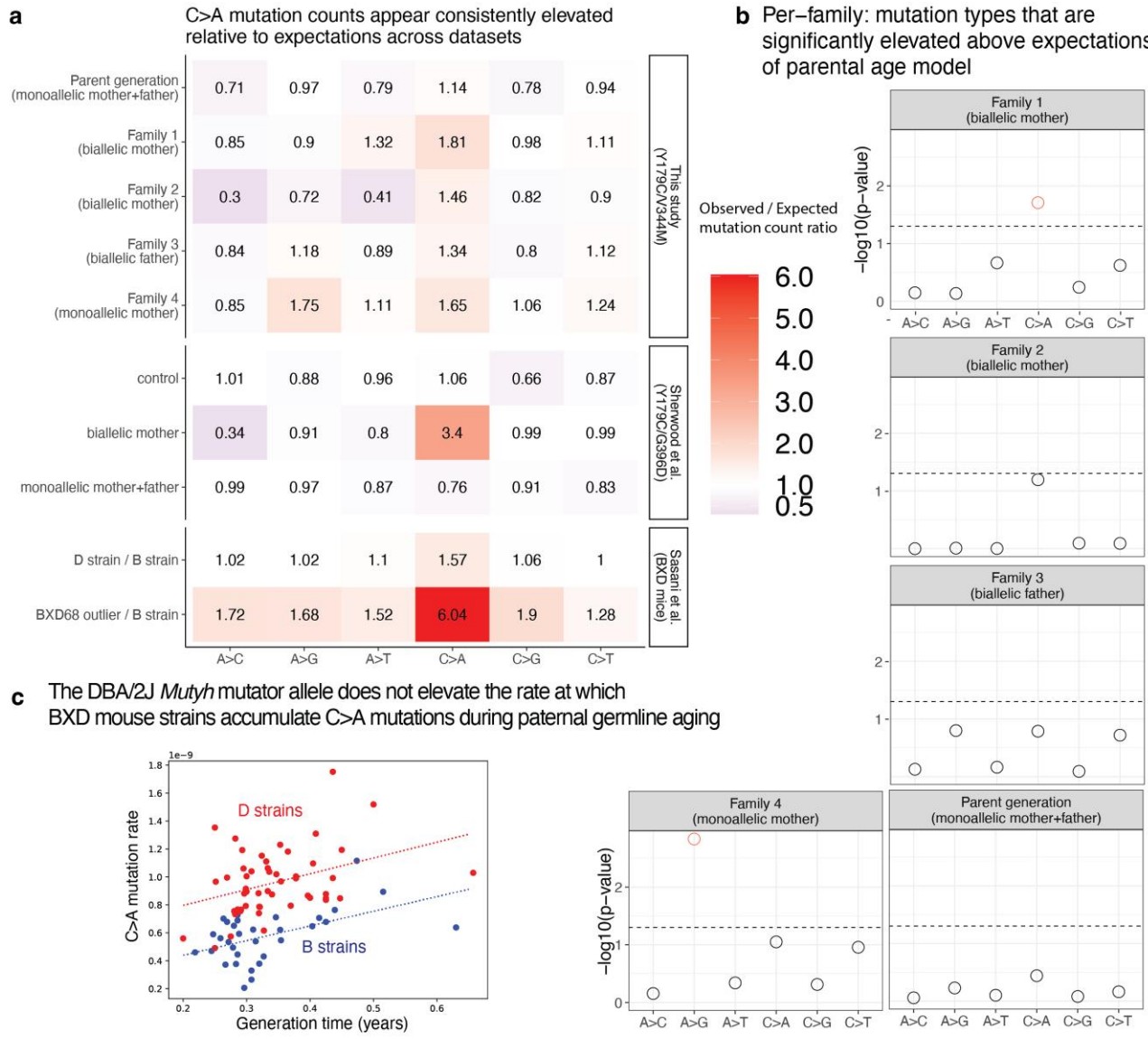

**Fig. 6.** Children of mothers with biallelic *MUTYH* genotypes show significantly elevated C>A DNM counts. a) A heat map showing the ratio of the observed/expected mutation counts per family (calculated by summing up the mutation counts per mutation type across all children within a family). These ratios are compared to the observed/expected ratio for the groups in Sherwood *et al.* (2023) (control group, individuals with a biallelic mother, and individuals with monoallelic parents), with expectations calculated using the parental age model. The bottom 2 rows show results from Sasani *et al.* (2022) for inbred BXD mouse strains: the "D" strain has an elevated mutation rate relative to the "B" strain, which has been linked to variation in *Mutyh*, and BXD68 is a mouse individual with an extreme outlier C>A mutator phenotype caused by a homozygous loss of function nonsynonymous mutation. The mouse ratios compare the per-generation rate of each mutation type between sets of inbred BXD mouse strains with different *Mutyh* genotypes. b) The probability of observing a mutation count of each of the six 1-mer mutation types under the parental age model that is greater than or equal to what we observed for each family in the pedigree. Points above the dashed line (red circles) fall below the upper 1-tailed Poisson $P < 0.05$ significance threshold. Family 1 shows significant elevation for C>A DNM counts above what is expected under the parental age model, and family 4 shows significant elevation of A>G mutations. c) Multilinear regression of C>A mutation rate per generation as a function of generation time across BXD mouse strains. The *Mutyh* allele affects the regression intercept but not the slope, implying that mutator strains and nonmutator strains accumulate C>A mutations at the same rate during parental ageing but accumulate these mutations at different rates during embryonic development.

We confirmed that our parental age model significance-testing framework was able to distill some of the main findings of Sherwood *et al.*'s (2023) study of germline mutator effects: in particular, the combined C>A burden of the children of the Sherwood *et al.* biallelic mother exceeded the 95% 1-tailed confidence interval of the parental age model (Supplementary Fig. 9). In addition, all children of *POLE* and *POLD1* variant carriers in Sherwood *et al.* (genotypes that appear to have much more severe germline mutator effects than *MUTYH*) significantly exceeded the C>A and A>G mutation burdens predicted under the parental age model

(Supplementary Fig. 9). We calculated a significant 3.4-fold enrichment of C>A mutations above the parental age model expectation in the family with a biallelic maternal *MUTYH* genotype sequenced by Sherwood *et al.* (Fig. 6a, Supplementary Fig. 11), suggesting that this family's p.Y179C/G368D genotype may have a more severe mutator phenotype than the p.Y179C/V234M genotype affecting our pedigree.

Unlike Sherwood *et al.* (2023), we did not detect a significant increase in overall DNMs phased to the haplotype of the carrier parent (Supplementary Figs. 12 and 13 and Table 1). However, we did

detect a significant elevation in C>A mutations phased to the maternal haplotype in Family 1 (1 of the 2 families in our pedigree where the mother is the biallelic *MUTYH* variant carrier) (Supplementary Fig. 14 and Table 3), indicating that there may be a carrier parent-specific elevation of C>A mutations in this family. We note that this result is based on very low sample sizes of phased DNMs: 3 C>A mutations phased to the maternal haplotype in family 1, compared to an expectation of 0.79 mutations, and so may be largely driven by stochasticity. In their biallelic families, Sherwood *et al.* were able to detect the activity of COSMIC mutational signature SBS18, a signature associated with defective *MUTYH* DNA repair (Alexandrov *et al.* 2020). However, mutational signature analysis of our DNM data did not identify any activity of either of the *MUTYH*-associated signature SBS18 or SBS36. This likely reflects the small total sample size of C>A mutations in our data (Supplementary Fig. 15) and should not be interpreted as evidence of absence of SBS18/SBS36.

### Estimating C>A mutator effect sizes in the maternal and paternal germline

One consistent feature of human germline mutagenesis is that only about 25% of mutations appear to arise in the maternal lineage. In a family where the mother's *MUTYH* genotype is pathogenic but the father's genotype is normal, any elevation of the C>A mutation rate observed in the children likely arises due to either excess mutations that arose in the oocyte prior to conception or due to postzygotic mutations. Even if a child has inherited a normal *MUTYH* allele from their father, their postzygotic mutations may still be enriched for C>A if they arise prior to the maternal–zygotic transition, when the embryo first begins to express paternally inherited genes.

Given that almost 75% of germline mutations originate in the spermatocytes, the minimum maternal C>A enrichment required to explain our data is expected to be larger than the overall C>A enrichments recorded in Fig. 6. To estimate the maternal germline C>A mutator effects that are required to explain the data, we started with the observed C>A mutation counts in Families 1 and 2 and subtracted the maternal and paternal C>A counts expected under the parental age model (Supplementary Fig. 16a). We then added each excess C>A count to the expected maternal C>A count and computed the proportional inflation of this value above the expected maternal C>A count. Using this logic, we calculated that the 1.46- to 1.81-fold overall C>A rate elevations observed in Families 1 and 2 (Fig. 6a, Supplementary Fig. 16a) imply maternal C>A mutation rate elevations of 4.2- and 2.7-fold, respectively (Supplementary Fig. 16b). The maternal effect implied by Sherwood *et al.*'s (2023) 3.4-fold increase in overall C>A count is even larger: this value translates to a 10.2-fold elevation of the maternal C>A mutation rate (Supplementary Fig. 16b).

Although Family 3's nonsignificant 1.34-fold C>A mutation rate elevation is not much lower than family 2's C>A enrichment, it implies a much lower paternal C>A rate elevation of only 1.45-fold (Supplementary Fig. 16b). This is very different from the 4.2- and 2.7-fold maternal C>A rate elevations that we infer to affect the 2 mothers who share the same biallelic *MUTYH* genotype. To investigate the likelihood that we were simply underpowered to detect a male germline mutator effect in family 3, we calculated a "mutator detection threshold" for each family, which is the minimum number of extra C>A mutations required to produce a significant deviation from the parental age model (horizontal black bars in Supplementary Fig. 16a).

We then calculated how much this minimum number of extra C>A mutations should inflate the germline rate in the parent with the biallelic *MUTYH* genotype: this is the minimum fold elevation of the biallelic parent's C>A mutation rate that we have power to detect (horizontal black bars in Supplementary Fig. 16b). Supplementary Fig. 16b compares these minimum effect sizes to the effect sizes estimated using our empirical data (orange points). According to these calculations, we should have power to detect a paternal C>A mutator effect of 1.8-fold or greater, which is notably smaller than the maternal effects supported by the data, yet exceeds the level of paternal C>A enrichment that is supported by the data. We also carried this analysis on a per-individual level, rather than summed per family (Supplementary Fig. 17). Although this analysis is based on a limited sample size of individuals, it suggests that *MUTYH* variants may have a proportionally stronger effect on the maternal germline and/or the early embryo compared to the paternal germline.

### Mutyh variation does not increase the strength of the paternal age effect in the BXD mice

To further investigate the etiology of *MUTYH*'s germline mutator effect, we turned our attention from the human genotype p.Y179C/V234M to the mutator allele affecting the murine homolog *Mutyh* in the mouse strain DBA/2J. This mutator allele, which we call *Mutyh-D*, occurs in half of the recombinant inbred mouse strains known as the BXDs, which are each descended from crosses of DBA/2J with the standard lab strain C57/BL6. Each BXD strain has been inbred for tens or hundreds of generations and was previously whole-genome sequenced, which allowed the average mutation rate over the inbreeding period to be measured with high precision (Sasani *et al.* 2022). These rates revealed that the "D strains," which have DBA/2J ancestry at *Mutyh*, have higher C>A mutation rates than the "B strains," which have C57/BL6 ancestry at this locus (Sasani *et al.* 2022). We were able to leverage these data to measure how the C>A mutation rate depends on parental age in the D strains as opposed to the B strains. Each BXD mouse strain has been inbred for a known number of generations spanning a known number of years, and we used these records to calculate each strain's average generation time. We found that these rates ranged from 0.2 to 0.63 years, spanning more than half of the mouse reproductive lifespan.

We fit a multilinear regression model to infer the dependence of the C>A mutation rate on *Mutyh* genotype (B or D) jointly with parental age, letting the *y*-intercept of the model be the C>A mutation rate at the minimum parental age of 0.2 years (Fig. 6c). We inferred a parental age effect of $1.05 \times 10^{-9}$ additional C>A mutations per site per year (ANCOVA $P < 0.001$), but found no significant interaction between this parental age effect and *Mutyh* genotype (ANCOVA $P > 0.82$), indicating that the rate of C>A mutations occurring during gamete ageing does not appear to differ between the B and D strains. In contrast, the baseline C>A mutation rate at age 0.2 years differs significantly between the B and D strains ($4.39 \times 10^{-10}$ vs $7.96 \times 10^{-10}$ mutations per site per generation; ANCOVA $P < 0.001$). This suggests that the elevated C>A mutation rate associated with the DBA/2J *Mutyh* allele is driven primarily by early embryonic mutations, not mutations that occur in the paternal (or maternal) gametes.

## Discussion

We have investigated the germline mutation rate and spectrum within a large extended family affected by a *MUTYH* genotype, p.Y179C/V234M, consisting of a relatively common pathogenic variant plus a rarer variant with conflicting interpretations. This family's history of colon cancer previously suggested that the

p.Y179C/V234M genotype had a pathogenic effect, and we were able to use a cell-based in vitro functional assay to classify p.V234M as a partial loss of function variant. By calling DNMs in the children of 2 mothers with the p.Y179C/V234M genotype, we documented a modest but significant maternal mutator effect that appears weaker than the maternal germline mutator effect recently discovered in the children of 2 mothers with the more common MAP-associated genotype p.Y179C/G368D (Sherwood et al. 2023). A complementary analysis of parental age dependence of the BXD mouse *Mutyh* mutator effect confirms that the mutator is unlikely to act on the paternal germline and is most likely to increase the mutation rate during embryonic development, similarly to a maternal mutator allele that Stendahl et al. (2023) recently discovered in rhesus macaques.

Even in a pedigree as large as the one we study here, DNM data sparsity limits the power to estimate precise mutator effect sizes. Based on prior knowledge about the biology of *MUTYH*, we expected to see excess germline C>A mutations in the children of biallelic carriers, and though our data appear to support this hypothesis, the observed C>A enrichments are likely not extreme enough to survive a stringent Bonferroni correction for the number of distinct tests performed throughout the manuscript, let alone an agnostic scan for mutators affecting other mutation types. We did not attempt to formulate a less conservative multiple test correction by estimating the number of truly independent tests being performed, which would have been challenging to do given the nested nature of testing both individuals and larger nuclear families for the same mutator effect. To give readers an accurate sense of data heterogeneity and noise, we perform more tests than the minimum number required, computing C>A enrichments individual by individual and observing nominally significant enrichments in only a few children ($P < 0.05$ in a 1-tailed test without multiple testing correction).

To our knowledge, this study is the first to call DNMs in the children of a father with a biallelic *MUTYH* genotype. Since about three-fourths of human variation arises in the paternal germline, we expected to have more power to measure a germline mutator effect in this family compared to families with maternal *MUTYH* variation. We were thus quite surprised that this father was the only biallelic parent whose children did not have a significantly elevated C>A mutation load, suggesting that *MUTYH* variation has a proportionally weaker effect on the paternal germline. This result should be interpreted with caution given our small sample sizes, but it could indicate that oxidative stress causes a smaller proportion of mutations in spermatocytes compared to oocytes or the developing embryo, or else that spermatocytes rely more on DNA repair pathways not involving *MUTYH*. Although it is possible that the one biallelic father in our data set is an outlier, our mouse analysis also fails to detect an effect of *Mutyh* variation on the spermatocytes, which would be expected to increase the parental age dependence of the C>A mutation rate, and points to the embryo as the most likely site of the mutator effect.

Maternal biallelic *MUTYH* genotypes could plausibly increase the rate of germline C>A mutations during the first few embryonic cell divisions prior to the maternal–zygotic transition; maternal DNA repair machinery is responsible for repairing embryonic DNA damage prior to the activation of zygotic transcription (Huang et al. 2014; Harland et al. 2017). Early embryonic mutations are enriched for C>A, possibly due to 8-OG damage that occurs during oocyte and spermatocyte maturation, and maternal *MUTYH* mutations may interfere with the repair of such damage (Smith, Dun, et al. 2013; Ohno et al. 2014; Gao et al. 2019). Advanced maternal age appears to increase the rate of C>A

mutations occurring on the paternal haplotype of the embryo (Gao et al. 2019), and biallelic maternal *MUTYH* mutations might cause a similar attenuation of 8-OG repair during the earliest stage of development.

One possibility is that 8-OG lesions cause similar absolute numbers of mutations per generation in males and females, but that the excess male mutation load is caused by factors unrelated to oxidative stress, which would seemingly contradict the widespread assertion that oxidative stress is a major cause of DNA damage in ageing sperm (Aitken et al. 2003; Aitken 2020; Aitken and Krausz 2001). Further study of germline mutagenesis in families with paternal *MUTYH* mutations may thus shed light on the etiology of germline mutagenesis in males with normal *MUTYH* genotypes, helping us better understand whether oxidative stress is truly to blame for age-related infertility and the genetic disorders associated with paternal age.

Because germline mutator phenotypes appear to be rare, at least at the current limits of our ability to detect them, these phenotypes have often been measured in the offspring of just one carrier parent, leaving us no information about whether these phenotypes are sex specific. The mutator phenotypes recently measured by Kaplanis et al. (2022) were mostly found to affect male parents, and a study of an extended family affected by a DNA polymerase delta mutator definitively measured a stronger effect in male carriers compared to female carriers (Andrianova et al. 2023). Though the *MBD4* mutator allele that was recently discovered in rhesus macaques clearly exerts a maternal effect, the absence of a male breeder with the same phenotype precluded estimation of the relative strength of the corresponding male mutator phenotype (Stendahl et al. 2023). Pedigree studies like ours and the work of Andrianova et al. (2023) will likely be instrumental for further study of possible sex differences affecting mutagenesis and DNA repair.

A technical innovation that improved the power of this study was new methodology for calling DNMs in incomplete nuclear families, with siblings acting as surrogate parents. Given our goal of calling DNMs in the children of individuals with rare pathogenic *MUTYH* genotypes, we were able to maximize our pool of study subjects by relaxing the usual restriction to calling DNMs only in children whose parents' genomes were both available for sequencing. Our surrogate parent approach does have drawbacks compared to DNM calling in complete nuclear families, most notably the restriction of the callable genome to regions where the proband inherited the same missing parental haplotypes as at least one sequenced sibling. Surrogate-based DNM calling is also susceptible to false positives caused by gene conversion and errors in IBD calling, and we note that these false positives will look cleanly mapped upon visual inspection of sequencing reads. The filters we employed to control these errors may have increased our false negative rates, particularly when using both a surrogate mother and a surrogate father. Conversely, our false positive rates appear to be elevated when only a single sibling is available as a surrogate parent and sib-sharing cannot be used to identify false positive DNMs that were actually inherited from the missing parent.

One new technology that will likely improve the performance of the surrogate method in the future is high-fidelity long-read sequencing, which enables nearly all mutations to be phased to their maternal or paternal haplotype of origin (Noyes et al. 2022; Porubsky et al. 2024). When comparing the genomes of siblings who were sequenced using PacBio HiFi or a similar technology, it will become very straightforward to determine whether any read from the proband is derived from the same haplotype as a read present in a parent or surrogate parent, which will further

limit the ability of inherited variants to masquerade as DNMs. Surrogate-based mutation calling has the potential to make DNM analysis accessible to the many families where parents are deceased or not in contact with their children, including families affected by rare or undiagnosed genetic diseases where DNM calling is likely to have the greatest scientific and clinical utility.

Our data suggest that the germline mutator effect of *MUTYH* predominantly operates in a recessive manner, paralleling its role in cancer predisposition. However, we note that all available data on C>A mutation rates in normal human cells are derived from individuals who have at least one loss of function allele (p.Y179C). Although our study and previous studies (Sherwood *et al.* 2023) find mutagenesis and cancer risk to be associated with biallelic genotypes that combine p.Y179C with a partial loss-of-function allele (p.V234M or p.G368D), we do not have similar data from biallelic genotypes that combine 2 partial loss of function alleles, and we still lack an estimate of the human germline effect of 2 complete loss of function alleles. As we move toward better quantification of partial loss-of-function genotypes, it will be important to consider how they interact epistatically with each other and additional genes—for example, variants that impair the function of *MUTYH* and *OGG1* appear to interact epistatically in both the germline and the soma (Robinson *et al.* 2022; Sasani *et al.* 2024).

The apparent effect size difference between p.Y179C/V234M and p.Y179C/G368D suggests that there may be utility in moving beyond the binary classification of *MUTYH* variants as simply pathogenic or nonpathogenic. Although data sparsity issues imply that this effect size difference should be interpreted with caution, recent studies of *MUTYH* mutator alleles in the mouse germline and the human soma have also found that some genotypes have more severe mutator phenotypes than others. Previous somatic mutation data found an effect size difference between the common genotypes p.Y179C/G368D and p.Y179C/Y179C that appeared concordant with an earlier age of polyposis onset in p.Y179C/Y179C carriers (Robinson *et al.* 2022). For a rare genotype like p.Y179C/V234M, epidemiological data can likely not predict variant effect severity, and sequencing of normal tissues obtained from carriers of this genotype may prove to be a more viable option for obtaining this information. In this way, the mutation load in healthy tissues like the germline might eventually prove useful for predicting the severity of cancer risk likely to be associated with different pathogenic *MUTYH* genotypes, allowing clinicians to use whole-genome sequencing to discern whether a family or an individual with a suspicious DNA repair variant is accumulating mutations in normal tissues faster than expected and might be at elevated risk of acquiring a mutation that transforms normal tissue into cancer.

## Data availability

All genomic data are available for controlled access via dbGaP, accession number phs003554.v1.p1. Per-individual de novo mutation counts and mutation spectra are available in Supplementary Tables 1–3. Custom scripts necessary for reproducing our analyses are available on GitHub at https://github.com/harrispopgen/mutyh_human_pedigree. A human reference panel of phased VCF files from the high-coverage 1000 Genomes Project (Byrska-Bishop *et al.* 2022) was used to phase the data and infer shared haplotype tracts between relatives. These data can be found at https://www.internationalgenome.org/data-portal/data-collection/30x-grch38. Poisson regression coefficients used for the parental age model can be found in Jónsson *et al.*'s (2017) Table S9. Sherwood *et al.*'s (2023)

de novo mutation counts and mutation spectra are found in Table 1 and Table S2 of that study, respectively.

Supplemental material available at GENETICS online.

## Acknowledgments

We thank all the study participants for their time and engagement with our research. We thank Martha Horike-Pyne for her assistance drafting consent forms and applying for Institutional Review Board approval, and we thank Jailanie Kaganovsky and Vidha Sudhesh for their assistance mailing DNA collection kits to the study participants. We also thank the editor and 3 anonymous reviewers for providing feedback that helped us improve the manuscript. We also benefited from helpful discussions with Rosana Risques, Lea Starita, and members of the Harris lab.

## Funding

The collection and sequencing of all human subject data were funded by a Searle Scholarship to KH by the Kinship Foundation. We acknowledge additional financial support from NIH NIGMS grant R35GM133428, a Burroughs Wellcome Fund Career Award at the Scientific Interface, a Pew Scholarship, the Allen Discovery Center for Cell Lineage Tracing, and a Sloan Fellowship, all to KH. ACB received additional support from the NIH NIA Biological Mechanisms of Healthy Aging training grant T32 AG066574, and CLY received support from the NIH NIGMS Cellular and Molecular Biology training grant T32 GM007270. SLH and JOK were supported by NIH NIGMS R01 GM129123. BHS received support from the Damon Runyon-Rachleff Innovation Award (DRR-33-15) and the Brotman Baty Institute for Precision Medicine.

## Conflicts of interest

BHS consults for the company Constantiam Biosciences. JOK serves as a scientific advisor to the company MyOme. The authors declare no other competing interests.

## Author contributions

KH conceived the study. BHS identified a suitable family for study, collected this family's genotype and phenotype data, and led the design of the human subject engagement and biological sample collection protocol. KH contacted and consented the study participants. CLY coordinated the sample processing and genomic data generation. ACB, DM-P, and KH designed the study's computational analysis framework. Computational analyses were carried out by CLY, ACB, DM-P, KH, and LZ. The cell line variant assay was designed by JOK and SLH and executed by SLH. CLY, ACB, KH, and SLH generated main text and SI figures. CLY, ACB, and KH drafted the manuscript, and DM-P, SLH, LZ, JOK, and BHS contributed manuscript edits.

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

*Editor: M. Hahn*