## [Peer Review File · Genetics]

A maternal germline mutator phenotype in a family affected by heritable colorectal cancer

Candice Young, Annabel Beichman, David Mas Ponte, Shelby Hemker, Luke Zhu, Jacob Kitman, Brian Shirts, and Kelley Harris

NOTE: The reviews and decision letters are unedited and appear as submitted by the reviewers. In extremely rare instances and as determined by a Senior Editor or the EIC, portions of a review may be redacted. If a review is signed, the reviewer has agreed to no longer remain anonymous. The review history appears in chronological order.

Review Timeline:

Submission Date:	2024-01-09
Editorial Decision:	2024-02-07
Resubmission Received:	2024-09-24
Editorial Decision:	2024-09-30
Resubmission Received:	2024-10-04
Accepted:	2024-10-08

February 5, 2024

GENETICS-2024-306776

A maternal germline mutator phenotype in a family affected by heritable colorectal cancer

Dear Kelley:

Two experts in the field have reviewed your manuscript, and I have read it as well. I think it is a solid study of an interesting mutation phenotype, with a seemingly more novel methodological contribution. While your manuscript is not currently acceptable for publication in GENETICS, we would welcome a substantially revised manuscript. Both reviewers have comments and concerns to be addressed in a revised manuscript. You can read their reviews at the end of this email.

I think the biggest barrier to publication brought up by the reviewers is the relative lack of novelty and power in the pedigree examined, especially as compared to previous studies of MUTYH. I agree with this. However, I was quite impressed by the "surrogate parent" approach to calling mutations, and think that a revision that brings this forward would be a substantial contribution to the field. In order to do this, a number of minor things would need to be changed (for instance, mentioning the approach in the Abstract), but I would also expect additional simulations or power calculations, as well as some more detail about the different approaches. I have added some comments of my own to the end of the letter along these lines.

We look forward to receiving your revised manuscript. Please let the editorial office know approximately how long you expect to need for revisions.

Upon resubmission, please include:

1. A clean version of your manuscript;
2. A marked version of your manuscript in which you highlight significant revisions carried out in response to the major points raised by the editor/reviewers (track changes is acceptable if preferred);
3. A detailed response to the editor's/reviewers' feedback and to the concerns listed above. Please reference line numbers in this response to aid the editor and reviewers.

Your paper will likely be sent back out for review.

Additionally, please ensure that your resubmission is formatted for GENETICS
<https://academic.oup.com/genetics/pages/general-instructions>

Follow this link to submit the revised manuscript: Link Not Available

Sincerely,

Matthew Hahn
Associate Editor
GENETICS

Approved by:
Jeff Sekelsky
Senior Editor
GENETICS

Reviewer #1 (Comments for the Authors (Required)):

This study focuses on a large, three-generation family pedigree affected by pathogenic MUTYH mutations, specifically the Y179C/V234M genotype. The researchers performed whole-genome sequencing on three siblings who were compound heterozygotes for these MUTYH variants and assessed their impact on germline mutation rates. Two of the biallelic siblings had a history of colon cancer, and others had colon polyps, indicating an elevated colorectal cancer risk in the family. Functional assays were conducted to evaluate the MUTYH variants' effects, revealing severe loss of repair function for Y179C and partial loss for V234M. The study also explored de novo mutation calling in nuclear families, especially in cases where the paternal genome was unavailable. Surrogate parent methods were employed to estimate germline mutation rates, revealing a modest but significant maternal mutator effect associated with C>A mutations, primarily in families with biallelic MUTYH mothers. The

study found a weaker or absent mutator effect in the paternal germline of a family with a biallelic MUTYH father.

Overall, this research provides insights into the germline mutation rates associated with the MUTYH Y179C/V234M genotype and suggests a female-biased mutator effect. The findings contribute to understanding the complex relationship between MUTYH genotypes, germline mutation rates, and colorectal cancer risk within a familial context.

Despite the commendable effort put into the work, there is limited novelty in the manuscript. Many of the concepts presented are more comprehensively addressed in previously published studies with stronger statistical power. Additionally, I have listed some of my major concerns below:

The study acknowledges that data sparsity limits the power to estimate precise mutator effect sizes. The small sample size, especially in families with biallelic MUTYH fathers, introduce variability and make it challenging to draw definitive conclusions.

The study suggests a potential difference in the mutator effect sizes between specific MUTYH genotypes (e.g., Y179C/V234M and Y179C/G368D). However, the effect size differences should be interpreted cautiously due to data sparsity and the possibility of variability in individual responses.

The study discusses a potential sex-specific effect, indicating a stronger mutator effect in the maternal germline compared to the paternal germline. The conclusion is based on a limited number of families, and further investigation with larger sample sizes is needed to confirm and understand potential sex-specific differences.

Mutational signature analysis did not identify activity of MUTYH-associated signatures SBS18 or SBS36. Inconsistencies in mutational signatures may be attributed to factors such as sample size, tissue specificity, or the limitations of current analytical methods.

The study involves variants classified differently in ClinVar (Y179C as pathogenic and V234M as a variant of uncertain significance). The variability in classification may impact the interpretation of results, and the study emphasises the need for functional assays to understand the pathogenicity of specific variants.

Addressing these issues and conducting further studies with larger sample sizes will contribute to a more robust understanding of MUTYH-related germline mutagenesis.

Reviewer #2 (Comments for the Authors (Required)):

Please see attachment.

Associate Editor Comments:

Major items:

- The approaches of Jonsson et al. 2018 (Nature Genetics) and Narasimhan et al. 2017 (Nature Communications) to identify mutations without information on the parental genotypes seem similar (but definitely not identical to) the surrogate approach used here, especially as applied to P1-P4. Both should be mentioned, as should any other similar approaches you are aware of.
- A figure that explains the approach used to call mutations in P1-P4 would help clarity, as would a clear divide (if there is one) between the different approaches used for different sampling designs.
- I am actually a bit unclear about how the maternal genomes are used when calling de novo mutations without a paternal sample. If all individuals have been phased using Beagle, what does the maternal information give you? As explained, the surrogate parent method does not need either parent. Maybe there is a step where you compare the maternal haplotype across the genome to the child's haplotype inherited from mom? Such a comparison (assuming phase was accurate) does not even need a sibling--Figure 2B implies that this is being done. Please clarify.
- A simulation that demonstrates the power of the surrogate method is warranted, and would help the paper to make its point. Such a simulation should probably also assess any error in the phasing step.
- Gene conversion in the unsampled parental germline is a concern for approaches like this (see Narasimhan et al. 2017). However, with more than two siblings it might be possible to check the third sibling for the alternative parental haplotype to see if the "mutant" allele is found there. Regardless, this and other assumptions of the method should be made clear.
- You may want to make a stand-alone section of the Results on the surrogate method.

Minor items:

- Figure 1: I don't actually see the families labeled anywhere, though I can figure out which is which from the main text.

- Figure S2A is drawn as if it is a workflow, but I can't see how de novo mutations were called in the siblings upstream of identifying the IBD tracts. Is this really how it was done?
- Is Figure S3 relevant to the surrogate method? It is cited in the main text as if it was.
- I think the individual IDs in the caption to Figure S4 are incorrect, as C32 is repeated.
- page 8: does "better performance" simply mean more coverage here?

Young and collaborators present an original and elegant manuscript describing germline de novo mutation (DNM) rates in a family with history of colon cancer. The family presents inheritance of a pathogenic MUTYH variant and another variant with conflicting interpretation of pathogenicity. First, authors use a functional assay to describe the activity of variants. Then, using germline whole genome sequencing, called DNMs of the participants. In the absence of the father, authors took a clever approach, using sibling's haplotype as surrogate father's genome. DNM rates are presented adjusted to parent's age, specific nucleotide change and ascendance of MUTYH mutational profile (mono/bi allelic; mother/ father). Not all individuals, even siblings from biallelic mother or father, share the same tendency of increased C>A DNM, characteristic of MUTYH deficiency. However, clustering nuclear families DNMs, results indicate that biallelic mothers (but not biallelic fathers) are associated with higher incidence of C>A DNMs, showing a strong female-based mutator phenotype. Data presented suggest that the mutator effect of deficient MUTYH operates in a recessive manner, as occurs in conditions such as MAP (MUTYH associated polyposis). This study is original, well delineated, with solid controls and interesting results and conclusions.

Some minor issues should be considered, as listed below.

1) MUTYH associated polyposis (MAP) is a condition that causes numerous polyps. These polyps may eventually develop into adenocarcinomas, causing elevated risk of colorectal cancer in biallelic deficient MUTYH individuals. In the text, although it describes the high prevalence of colon cancer among members of this family, it is not clear whether individuals are diagnosed with MAP. This question is of higher concern checking the heredogram, in which there is not any individual presenting with polyposis AND colon cancer.

2) The description of high proportion of biallelic siblings carrying both variants (5 out of 7) as it is in the text puts into perspective the possibility of the variants to be present in cis. Checking the heredogram, it is clear that this is not the case, but the text could include which variant is of maternal and paternal inheritance.

3) According to the text: *"Two of the three biallelic siblings were previously diagnosed with colon cancer, while the other two siblings (as well as two additional biallelic siblings who did not participate in the study) had histories of colon polyps."*

Depending on the age, polyps, per se, is not a rare condition. The description of the family could be enriched including the age of participants and an estimated number of polyps or, preferably, if there is a clear diagnostic of MAP.

4) Considering MAP is an autosomal recessive condition, this family stands out by presenting three generations of affected members, presenting with colon cancer: 2/7 siblings; mother,

maternal aunt and uncle; and maternal grandmother. Considering the WGS information, it is important to rule out other (and more common) colorectal cancer predisposing conditions such as Lynch (autosomal dominant). This hypothesis is specially considered since (in the heredogram), individuals marked as “colon cancer” and for polyposis are mutually exclusive. Is there any information about MSI status in colorectal cancer cases among family members?

5) It would be informative to include the age of the participants, at least the age at cancer diagnosis.

Subtitle Figure 1: “*MUTYH* mutations and age at cancer diagnosis / number of identified colon polyps are listed below individuals studied in this pedigree.”

Information about age at cancer diagnosis is missing.

6) Saliva, the material used to obtain genomic DNA, is composed mainly of white blood cells. A hypothesis for explaining the high proportion of somatic mutations identified in P2, a patient treated for colon cancer, is clonal hematopoiesis. Are those somatic mutations enhanced for C>A mutations (COSMIC Sig 18, due to *MUTYH*) or for T>G mutations (~COSMIC Sig 17, due to 5-FU) (Christensen, 2019, Nat Communication - doi.org/10.1038/s41467-019-12594-8)?

7) It seems to be a discrepancy between Figure 4 A and 4B. Figure 4B shows increased C>A mutations only in families “1” and “2” both reporting biallelic mothers. However, Figure 4A, the heatmap showing the observed/expected ratios, C>A mutations in family 4 is very similar (and higher) than family 2. Please explain.

8) “In each case, the extracted signature was deconvoluted into signatures SBS1 and SBS5, two clock-like signatures that generally make up the bulk of mutations in both germline and somatic data. No contributions of SBS18 or SBS36, somatic mutational signatures associated with defective *MUTYH*, were detected.”

Consider running SigProfiler using COSMIC version 2 (Alexandrov 2013) instead of version 3 (Alexandrov 2020). *MUTYH* deficiency was initially associated with Signature 18 (Pilati 2017, Alexandrov is one of the authors – DOI: 10.1002/path.4880). Pilati report *MUTYH* deficiency in tumors: “Mutational signature analysis identified three distinct signatures closely related to the previously described signatures 1, 5, and 18”.

Another study associated *MUTYH* deficiency with Sig18 but did not achieve the same correlation using only SBS18 nor SBS36. However, the correlation obtained with Sig18 vs (SBS18+SBS36) was very high (Barreiro, 2022 - DOI: 10.1002/path.5829).

We thank the editor and the anonymous reviewers for their careful attention to our manuscript, which has helped us improve our presentation and analysis substantially. As requested by the editor, our revised manuscript more prominently features the novel methodology that we developed in order to call de novo mutations in incomplete nuclear families. We now use simulated data to demonstrate the power and accuracy of this approach, and we devote a standalone section of the results to an explanation of surrogate parent de novo mutation calling. We believe we have addressed all minor concerns about presentation and clarity, and we have made an effort to better situate our findings within the landscape of prior research. Although we acknowledge that some of the novelty of our manuscript was superseded by a study published a few months before we submitted ours, we note that *Genetics's* "No Scoop" policy indicates that the journal values independent corroboration of recently published results (<https://academic.oup.com/genetics/pages/general-instructions>).

In addition to addressing all direct requests for new analyses, we added an additional analysis that bolsters our claim that *MUTYH's* germline mutator effect is driven by the maternal germline. By reanalyzing published data from mouse lines affected by a mutator allele of the homologous gene *Mutyh*, we were able to triangulate the most likely site of the mutator activity to the early embryo, where DNA repair is carried out by maternal factors. We believe that this new analysis helps address reviewers' concern that our conclusions might not be robust due to our human dataset's relatively small sample size.

Associate Editor Comments:

I think the biggest barrier to publication brought up by the reviewers is the relative lack of novelty and power in the pedigree examined, especially as compared to previous studies of *MUTYH*. I agree with this. However, I was quite impressed by the "surrogate parent" approach to calling mutations, and think that a revision that brings this forward would be a substantial contribution to the field. In order to do this, a number of minor things would need to be changed (for instance, mentioning the approach in the Abstract), but I would also expect additional simulations or power calculations, as well as some more detail about the different approaches. I have added some comments of my own to the end of the letter along these lines.

Thank you for this vote of confidence and for these useful suggestions. We now mention the surrogate parent approach in the abstract as follows:

By developing novel methodology that uses siblings as "surrogate parents" to identify de novo mutations, we were able to include mutation data from several children whose parents were unavailable for sequencing. (Lines*** 28-30)

*****Note: the lines we refer to throughout this response are associated with the “clean” version of our manuscript, with no tracked changes included.**

We have also added a new section to the main text that includes a surrogate parent analysis of simulated data, showing how the power and precision of our mutation calling increases with the inclusion of more siblings. More detail on these revisions is given in response to your specific comments.

A new addition to the manuscript which further bolsters the novelty and utility of the surrogate method is the insight that a full sibling can actually act as another sibling’s “double surrogate parent” across the ~25% of the genome where the two siblings inherited the same maternal haplotype and also the same paternal haplotype (this is diagrammed in Figure 3C of the revised manuscript). By adding in these double surrogate mutation calls, we were able to modestly increase the proportion of the parent genomes (P1 through P4) which are accessible for DNM calling.

Major items:

-The approaches of Jonsson et al. 2018 (Nature Genetics) and Narasimhan et al. 2017 (Nature Communications) to identify mutations without information on the parental genotypes seem similar (but definitely not identical to) the surrogate approach used here, especially as applied to P1-P4. Both should be mentioned, as should any other similar approaches you are aware of.

We agree that our surrogate method builds on some important earlier advances, including Narasimhan, et al. 2017 as well as Tian, et al. 2019 and Tian, et al. 2022. Unless we’re mistaken, Jonsson, et al. 2018 actually has quite a different focus, since it uses two parents to call de novo mutations inherited by multiple children within each large family, though we cite this paper in support of our claim that at most 1-2% of DNMs are shared by multiple siblings. We now discuss this prior literature in lines 194–200 as follows:

Our use of the term “surrogate parent” is loosely based on the established use of relatives as surrogate parents for haplotype phasing (Kong et al. 2008). We also drew inspiration from several recent studies that successfully estimated mutation rates using tracts inherited identical-by-descent (IBD) between relatively distant relatives (Narasimhan et al. 2017; Tian et al. 2019, 2022). In contrast to these earlier studies, which estimated population-averaged mutation rates using mutations that likely occurred many generations ago, our method is designed to estimate a single individual’s mutation rate using mutations that arose in the contemporary individuals being sequenced.

-A figure that explains the approach used to call mutations in P1-P4 would help clarity, as would a clear divide (if there is one) between the different approaches used for different sampling designs.

This is a great suggestion. This is now Figure 3 in the revised manuscript:

Figure 3. Calling DNMs using only surrogate parents. **A)** Generation III of our pedigree contains four full siblings (P1, P2, P3, P4) whose parents are deceased. We were able to call DNMs in these individuals using siblings as surrogate mothers and fathers. Example regions of paternal and maternal haplotype sharing are regions where P2 and P4 can act as surrogate father and mother to P3. Red crosshatches denote parental haplotype segments that were inherited by exactly one sibling—these regions are inaccessible to DNM calling due to lack of a surrogate parent. **B)** Illustration of maternal and paternal haplotype sharing in two example surrogate trios. DNMs are callable within orange regions due to haplotype sharing with both the surrogate mother and the surrogate father. **C)** Illustration of the regions that are accessible for calling in P3 using P2 as a double surrogate parent. This is possible when the surrogate and the proband share both maternal and paternal haplotypes.

-I am actually a bit unclear about how the maternal genomes are used when calling de novo mutations without a paternal sample. If all individuals have been phased using Beagle, what does the maternal information give you? As explained, the surrogate parent method does not need either parent. Maybe there is a step where you compare the maternal haplotype across the genome to the child's haplotype inherited from mom?

Such a comparison (assuming phase was accurate) does not even need a sibling-- Figure 2B implies that this is being done. Please clarify.

To call DNMs in a particular region of a child's genome, we need to compare the child to individuals who inherited both the child's maternal and paternal haplotypes identical by descent within 1 generation. Then if we find an allele that is unique to the proband, we know it must have arisen de novo within one generation (or be some kind of bioinformatic artifact). So if the maternal genome is available, its sequence tells us which alleles are available to have been inherited from mom at each site in the genome and which are inconsistent with maternal inheritance. If the maternal genome is not available, then we need a sibling who shares maternal IBD, since sequence differences between the two siblings within the maternal IBD region are the putative locations of DNMs. We have added text to clarify this point in lines 188–192:

Instead of comparing each child's genome to their mother and father to identify variants that must have arisen de novo, we compared each child from Families 2 and 4 to their mother plus one or more siblings who inherited some of the same paternal DNA. If the child's genome contains a variant that is absent from both the mother and the sibling surrogate father, this implies that the unique variant arose de novo in the child (**Figure 2A-2B**; pipeline described in **Figure S2**).

The drawback of using a sibling as a surrogate mother is that two siblings only share the same maternal haplotype about 50% of the time, so this cuts down on the amount of genomic real estate where DNM calling is possible. With multiple siblings who can act as surrogate mothers and fathers, we can call DNMs across the majority of the genome, but real parents are preferable if available since they afford 100% genome coverage and avoid the potential for parental gene conversion to cause false DNM calls. This point is now clarified in lines 221–254 and demonstrated in **Figure 4A**:

In a large family with a mother and $n+1$ siblings, the only regions inaccessible for surrogate-father DNM calling will be regions where n siblings inherited the same paternal haplotype and the remaining sibling inherited the other paternal haplotype. In this scenario, the sibling who inherited the unique paternal haplotype has no one to serve as a surrogate father, so that sibling's genome will be inaccessible for DNM calling while the n other siblings' genomes will be accessible. If we consider the paternal haplotype inherited at a specific locus in a specific sibling's genome, the probability that any given sibling inherited the same paternal haplotype is $\frac{1}{2}$. Therefore, the probability that all n other siblings inherited the other paternal haplotype is 2^{-n} . If any one other sibling did inherit the same paternal haplotype, the first sibling's genome will be

accessible for DNM calling at the locus we are considering. This implies that in our hypothetical family with one parent and $n+1$ children, a fraction $1 - 2^{-n}$ of all DNMs should be callable. For example, in Family 2, which consists of 3 children and their mother, $\frac{3}{4}$ of all DNMs should be callable.

Since full siblings share DNA inherited from both of their parents, they can serve as surrogate parents for DNM calling even when the mother and father are both deceased, as is the case for the Generation III siblings P1, P2, P3, and P4 (for more details, see methods and **Figure 3A**). Since maternal and paternal chromosomes are passed down independently of one another, the fraction of DNMs accessible for calling in a family with $n+1$ children and no parents is expected to be $(1 - 2^{-n})^2$, slightly smaller than the fraction $1 - 2^{-n}$ that is callable when paternal or maternal DNA is available.

-A simulation that demonstrates the power of the surrogate method is warranted, and would help the paper to make its point. Such a simulation should probably also assess any error in the phasing step.

We now include an analysis of a simulated family containing a total of 5 siblings (lines 276-289 of the results and lines 905-975 of the methods). We analyze this family using our entire pipeline, including the computational phasing step, and Figure 4 compares the performance of standard DNM calling to surrogate calling using different configurations of real and surrogate parents. We did not explicitly address phasing error, but we note that phasing switch errors should not necessarily affect our performance—when we calculate the set of regions of maternal IBD between two siblings, we merge together IBD segments that seem to jump from one haplotype to another. IBD calling errors are more problematic to us since these would cause us to try to call mutations in regions where we do not in fact have good surrogate parents to use, but our density filter method is designed to detect and remove the resulting errors, and we tested the performance of this method on the simulated data along with the rest of the pipeline.

Analyzing this simulated dataset prompted us to make some improvements to our pipeline, including increasing the stringency of our filtering of clusters of putative de novo mutations that include at least one SNP that has been assigned rsIDs. Such SNPs might be true “repeat mutations” that occur at sites with preexisting population variation, but they may also indicate that germline variants are masquerading as de novo mutations due to surrogate tract calling errors. In our real data analysis, this pipeline change caused Family 2 to fall just below genome-wide significance when we test for an

elevated C>A mutation rate, though it still remains true that the only children with elevated C>A mutation rates are children of maternal *MUTYH* carriers (from Families 1 and 2).

-Gene conversion in the unsampled parental germline is a concern for approaches like this (see Narasimhan et al. 2017). However, with more than two siblings it might be possible to check the third sibling for the alternative parental haplotype to see if the "mutant" allele is found there. Regardless, this and other assumptions of the method should be made clear.

This is a great point. Lines 210-219 now acknowledge that paternal gene conversion can create errors and explain that we mitigate these errors by excluding common variants that appear to be DNMs:

In this setup, gene conversion between the two paternal haplotypes has the potential to create false positive DNMs, as previously observed by Narasimhan et al. (2017). To minimize these errors, we filtered out putative DNMs that were present in the 1000 Genomes data or in two or more members of our pedigree (this should eliminate gene conversion errors at any loci where not all siblings inherited the same paternal haplotype). We note that the familial mutation sharing filter will cause us to miss the 1-2% of DNMs that are shared between siblings due to germline mosaicism (Jónsson et al. 2018). The 1000 Genomes filter may also cause us to exclude some true DNMs, particularly at CpG sites or other mutation hotspots, but it should effectively filter out many false positive DNMs that were actually inherited from a missing parent, except when those inherited variants are very rare in the population as a whole.

-You may want to make a stand-alone section of the Results on the surrogate method.

We have reorganized the manuscript to include the relevant results in a section titled "*Development, application, and benchmarking of a method using siblings as "surrogate parents" for de novo mutation calling.*"

Minor items:

-Figure 1: I don't actually see the families labeled anywhere, though I can figure out which is which from the main text.

Great point, we have added these family labels to the pedigree as shown below (reproduced from Figure 1).

-Figure S2A is drawn as if it is a workflow, but I can't see how de novo mutations were called in the siblings upstream of identifying the IBD tracts. Is this really how it was done?

This is in fact how the calling was done, since phasing and IBD information are not part of the input to the DNM caller we used (GATK possible de novo). The IBD information is then used to filter out large swaths of these IBD calls that occur in regions where the siblings being used as surrogate mothers and/or fathers are not in fact valid surrogate parents. In addition, the preliminary DNM calls are used to filter out regions where IBD appears to have been called incorrectly. If the density of DNM calls is extremely high, this is most parsimoniously explained by an error in the surrogate parent accessibility mask, so we adjust the mask to exclude these regions (see “Filtering” in **Methods** and **Figure S2B**). We have rephrased the text in the first workflow box to “Calling candidate DNMs” to emphasize that these calls still need to go through filtering later.

Updated figure:

-Is Figure S3 relevant to the surrogate method? It is cited in the main text as if it was.

We have deleted this reference since it is now superseded by new figures that better make the point about error correction that we were obliquely trying to make.

-I think the individual IDs in the caption to Figure S4 are incorrect, as C32 is repeated.

Thanks, this typo has been corrected.

-page 8: does "better performance" simply mean more coverage here?

Here, "better performance" also refers to the use of redundant surrogate parents for error correction of the DNM calls. This discussion has been significantly revised due to the addition of the benchmarking simulation, and lines 281-284 now discuss the error correction idea:

As illustrated in **Figure 4B**, different sets of surrogate parents provided coverage of complementary genomic regions, and overlap between these tracts permitted error-correction of false positives that only appear when specific sets of surrogate parents are used.

Reviewer #1 (Comments for the Authors (Required)):

This study focuses on a large, three-generation family pedigree affected by pathogenic MUTYH mutations, specifically the Y179C/V234M genotype. The researchers performed whole-genome sequencing on three siblings who were compound heterozygotes for these MUTYH variants and assessed their impact on germline mutation rates. Two of the biallelic siblings had a history of colon cancer, and others had colon polyps, indicating an elevated colorectal cancer risk in the family. Functional assays were conducted to evaluate the MUTYH variants' effects, revealing severe loss of repair function for Y179C and partial loss for V234M. The study also explored de novo mutation calling in nuclear families, especially in cases where the paternal genome was unavailable. Surrogate parent methods were employed to estimate germline mutation rates, revealing a modest but significant maternal mutator effect associated with C>A mutations, primarily in families with biallelic MUTYH mothers. The study found a weaker or absent mutator effect in the paternal germline of a family with a biallelic MUTYH father.

Overall, this research provides insights into the germline mutation rates associated with the MUTYH Y179C/V234M genotype and suggests a female-biased mutator effect. The findings contribute to understanding the complex relationship between MUTYH genotypes, germline mutation rates, and colorectal cancer risk within a familial context.

Despite the commendable effort put into the work, there is limited novelty in the manuscript. Many of the concepts presented are more comprehensively addressed in previously published studies with stronger statistical power.

We agree that some of the novelty of our results was superseded by an excellent study that was published by Sherwood, et al. in *Nature Communications* a few months before we submitted our paper. However, we were in part motivated to submit our paper to *Genetics* because of the “No scoop policy” featured on the website’s instructions to authors, which states:

“Because the editors appreciate that competing studies often complement each other, recent publication of similar articles by others does not necessarily preclude consideration of a manuscript for publication in GENETICS.”

In our revised manuscript, we have made an effort to highlight the things that distinguish our study from Sherwood, et al. including the surrogate parent DNM calling method, the analysis of both female and male carriers of pathogenic *MUTYH* genotypes, and a rigorous model for inferring mutator effects against a background model that accounts for parental age effects.

Although our sample sizes are not large, we argue that they are large enough to provide compelling evidence for a sex difference in the germline effect of this family's *MUTYH* genotype, which was not apparent in any previous studies and sheds light on a difference between the DNA damage and repair processes that are experienced by sperm and oocytes. We also distinguish ourselves from Sherwood by developing a statistical framework for quantifying how much each individual and each family departs from a null model based on the mutation rates and spectra of healthy families spanning a variety of paternal and maternal ages. In our opinion, this framework not only showcases the subtle signal present in our own data, but also makes it easier to interpret the striking patterns in the Sherwood data, as shown in **Figure S8**.

Additionally, I have listed some of my major concerns below:

The study acknowledges that data sparsity limits the power to estimate precise mutator effect sizes. The small sample size, especially in families with biallelic *MUTYH* fathers, introduce variability and make it challenging to draw definitive conclusions.

We agree that it isn't possible to rule out the idea that another biallelic *MUTYH* father might look different from the one in our dataset. However, as we noted in our statistical analysis, paternal mutators should in principle be easier to detect than maternal mutators due to the higher baseline burden of paternal mutations, as illustrated by the differences between the C>A enrichment values and detection thresholds in **Figure 6**. In addition, we were able to strengthen our argument that the mutator effect is maternal-specific by adding a new analysis of the generation-time dependence of a germline *Mutyh* mutator effect that was previously observed in mice. In mice, we show that *Mutyh* variation has little to no effect on the mutation rate in mature spermatocytes, but instead affects the rate of germline mutations occurring during early embryonic development. Prior to the maternal-zygotic transition, maternal factors are responsible for DNA repair in the developing embryo, meaning that an embryonic effect is consistent with the maternal inheritance pattern we see in humans. We describe this result as follows (lines 492-517):

Mutyh variation does not increase the strength of the paternal age effect in the BXD mice

To further investigate the etiology of *MUTYH*'s germline mutator effect, we turned our attention from the human genotype Y179C/V234M to the mutator allele affecting the murine homolog *Mutyh* in the mouse strain DBA/2J. This mutator allele, *Mutyh-D*, occurs in half of the recombinant inbred mouse strains known as the BXDs, which are each descended from crosses of DBA/2J with the standard lab strain C57/BL6. Each BXD strain has been inbred for tens or hundreds of generations and was previously whole-genome sequenced, which allowed the average mutation rate over the inbreeding period to be measured with high precision (Sasani et al. 2022). These rates revealed that the “D strains,” which have DBA/2J ancestry at *Mutyh*, have higher C>A mutation rates than the “B strains” which have C57/BL6 ancestry at this locus (Sasani et al. 2022). We were able to leverage these mutation rate estimates to measure how the C>A mutation rate depends on parental age in the D strains as opposed to the B strains. Each BXD mouse strain has been inbred for a known number of generations spanning a known number of years, and we used these records to calculate each strain's average generation time. We found that these rates ranged from 0.2 years to 0.63 years, spanning more than half of the mouse reproductive lifespan.

We fit a multilinear regression model to infer the dependence of the C>A mutation rate on *Mutyh* genotype (B or D) jointly with parental age, letting the y intercept of the model be the C>A mutation rate at the minimum parental age of 0.2 years (**Figure 6C**). We inferred a parental age effect of 1.05×10^{-9} additional C>A mutations per site per year (ANCOVA $p < 0.001$), but found no significant interaction between this parental age effect and *Mutyh* genotype (ANCOVA $p > 0.82$), indicating that the rate of C>A mutations occurring during gamete aging does not differ between the B and D strains. In contrast, the baseline C>A mutation rate at age 0.2 years differs significantly between the B and D strains (4.39×10^{-10} versus 7.96×10^{-10} mutations per site per generation; ANCOVA $p < 0.001$). This suggests that the elevated C>A mutation rate associated with the DBA/2J *Mutyh* allele is driven primarily by early embryonic mutations, not mutations that occur in the paternal (or maternal) gametes.

Figure 6C) Multilinear regression of C>A mutation rate per generation as a function of generation time across BXD mouse strains. The *Mutyh* allele affects the regression intercept but not the slope, implying that mutator strains and non-mutator strains accumulate C>A mutations at the same rate during parental aging but accumulate these mutations at different rates during embryonic development

The study suggests a potential difference in the mutator effect sizes between specific *MUTYH* genotypes (e.g., Y179C/V234M and Y179C/G368D). However, the effect size differences should be interpreted cautiously due to data sparsity and the possibility of variability in individual responses.

We agree with this, and advocate for a cautious interpretation of the effect size difference in lines 630–637:

The apparent effect size difference between Y179C/V234M and Y179C/G368D suggests that there may be utility in moving beyond the binary classification of *MUTYH* variants as simply pathogenic or non-pathogenic. Although data sparsity issues imply that this effect size difference should be interpreted with caution, recent studies of *MUTYH* mutator alleles in the mouse germline and the human soma have also found that some genotypes have more severe mutator

phenotypes than others. Previous somatic mutation data found an effect size difference between the common genotypes Y179C/G368D and Y179C/Y179C that appeared concordant with an earlier age of polyposis onset in Y179C/Y179C carriers (Robinson et al. 2022).

The study discusses a potential sex-specific effect, indicating a stronger mutator effect in the maternal germline compared to the paternal germline. The conclusion is based on a limited number of families, and further investigation with larger sample sizes is needed to confirm and understand potential sex-specific differences.

We agree that further investigation in larger cohorts would be valuable. Although we were not able to expand our human cohort, we were able to perform additional validation using a published mouse model, as described above. In addition, we would argue that part of the value of our study is validation of the *MUTYH* mutator effect found by Sherwood et al., which included only two families with biallelic *MUTYH* mutations, neither of which had a biallelic father as the carrier. So far, all studies of human germline mutator phenotypes have had fairly small sample sizes because these phenotypes are so rare. To our knowledge, the largest sample sizes of human mutator families to date have been Kaplanis, et al. 2022, with 12 families that all have different mutator phenotypes, and Robinson, et al. 2021 which sampled 14 families affected by *POLE* mutations. Due to the logistical challenges associated with compiling large cohorts of families affected by these rare phenotypes, we would argue that small studies that couch their results with appropriate caution are needed to move this field forward. We now highlight the similarly small sample size of the Sherwood study in lines 526-529:

By calling de novo mutations in the children of two mothers who carry the Y179C/V234M genotype, we documented a modest but significant maternal mutator effect that appears weaker than the maternal germline mutator effect recently discovered in the children of two mothers with the more common MAP-associated genotype Y179C/G368D (Sherwood et al. 2023).

Mutational signature analysis did not identify activity of *MUTYH*-associated signatures SBS18 or SBS36. Inconsistencies in mutational signatures may be attributed to factors such as sample size, tissue specificity, or the limitations of current analytical methods.

We agree with this. We clarify in lines 443-446 that the absence of these signatures is likely a power issue:

However, mutational signature analysis of our DNM data did not identify any activity of either of the *MUTYH*-associated signatures SBS18 or SBS36. This likely reflects the small total sample

size of C>A mutations in our data (**Figure S15**) and should not be interpreted as evidence of absence of SBS18/SBS36.

The study involves variants classified differently in ClinVar (Y179C as pathogenic and V234M as a variant of uncertain significance). The variability in classification may impact the interpretation of results, and the study emphasizes the need for functional assays to understand the pathogenicity of specific variants.

We agree that the annotation of V234M as “uncertain significance” is relevant to the interpretation of our results. This was part of our motivation for performing a functional assay of V234M, which we agree made the phenotypic difference between our families and the Sherwood families more interpretable than it otherwise would have been.

Addressing these issues and conducting further studies with larger sample sizes will contribute to a more robust understanding of MUTYH-related germline mutagenesis.

We agree that larger sample sizes would add a lot to our understanding of MUTYH's germline phenotype, but this would require funding to hire a clinical coordinator and do much larger-scale recontacting of people who share the same rare genotype. Proof of concept studies like ours are needed to lay the groundwork for obtaining the funding needed for larger-scale clinical studies of the relationship between rare cancer syndromes and germline mutation rate.

Reviewer #2 (Comments for the Authors (Required)):

Young and collaborators present an original and elegant manuscript describing germline de novo mutation (DNM) rates in a family with history of colon cancer. The family presents inheritance of a pathogenic MUTYH variant and another variant with conflicting interpretation of pathogenicity. First, authors use a functional assay to describe the activity of variants. Then, using germline whole genome sequencing, called DNMs of the participants. In the absence of the father, authors took a clever approach, using sibling's haplotype as surrogate father's genome. DNM rates are presented adjusted to parent's age, specific nucleotide change and ascendance of MUTYH mutational profile (mono/bi allelic; mother/ father). Not all individuals, even siblings from biallelic mother or father, share the same tendency of increased C>A DNM, characteristic of MUTYH deficiency. However, clustering nuclear families DNMs, results indicate that biallelic mothers (but not biallelic fathers) are associated with higher incidence of C>A DNMs, showing a strong female-based mutator phenotype. Data presented suggest that the mutator effect of deficient MUTYH operates in a recessive manner, as occurs in

conditions such as MAP (MUTYH associated polyposis). This study is original, well delineated, with solid controls and interesting results and conclusions.

Thank you for this comprehensive and positive assessment!

Some minor issues should be considered, as listed below.

1) MUTYH associated polyposis (MAP) is a condition that causes numerous polyps. These polyps may eventually develop into adenocarcinomas, causing elevated risk of colorectal cancer in biallelic deficient MUTYH individuals. In the text, although it describes the high prevalence of colon cancer among members of this family, it is not clear whether individuals are diagnosed with MAP. This question is of higher concern checking the heredogram, in which there is not any individual presenting with polyposis AND colon cancer.

We apologize for the ambiguity here. Since the diagnosis of MAP is based on genotype rather than phenotype (<https://www.ncbi.nlm.nih.gov/books/NBK107219/>), we can say that by virtue of their genotypes, the biallelic individuals in this family are affected by MAP. To avoid confusion, we state in lines 81–82 that

“*MUTYH*-associated polyposis (MAP) follows a recessive inheritance pattern, occurring in “biallelic” individuals who have inherited two sub-functional copies of the *MUTYH* gene (Morak et al. 2014).”

and we state in lines 96–97 that:

“...we sequenced fifteen genomes from a large extended family containing multiple individuals affected by MAP.”

2) The description of high proportion of biallelic siblings carrying both variants (5 out of 7) as it is in the text puts into perspective the possibility of the variants to be present in cis. Checking the heredogram, it is clear that this is not the case, but the text could include which variant is of maternal and paternal inheritance.

This is a good point, we now state in line 116 that “These siblings inherited Y179C from their father and inherited V234M from their mother.”

3) According to the text: *“Two of the three biallelic siblings were previously diagnosed with colon cancer, while the other two siblings (as well as two additional biallelic siblings who did not participate in the study) had histories of colon polyps.”*

Depending on the age, polyps, per se, is not a rare condition. The description of the family could be enriched including the age of participants and an estimated number of polyps or, preferably, if there is a clear diagnostic of MAP.

Our consent forms unfortunately don't permit us to share additional clinical detail about the study participants, but the diagnosis of MAP is based on genotype and not number of polyps, so we now state in line 118-119 that "all three [biallelic siblings] meet the diagnostic criterion for MAP by virtue of their *MUTYH* genotype".

4) Considering MAP is an autosomal recessive condition, this family stands out by presenting three generations of affected members, presenting with colon cancer: 2/7 siblings; mother, maternal aunt and uncle; and maternal grandmother. Considering the WGS information, it is important to rule out other (and more common) colorectal cancer predisposing conditions such as Lynch (autosomal dominant). This hypothesis is specially considered since (in the heredogram), individuals marked as "colon cancer" and for polyposis are mutually exclusive. Is there any information about MSI status in colorectal cancer cases among family members?

Our consent forms unfortunately do not permit sharing information about MSI status or anything else from a somatic pathology report, but we now report in lines 123-125 that "All individuals in the family tested negative for Lynch syndrome variants and other variants known to cause heritable colorectal cancer."

5) It would be informative to include the age of the participants, at least the age at cancer diagnosis.

Subtitle Figure 1: "*MUTYH* mutations and **age at cancer diagnosis** / number of identified colon polyps are listed below individuals studied in this pedigree."

Information about age at cancer diagnosis is missing.

We are unfortunately not able to include the participants' birth years or ages at cancer diagnosis since this is considered protected health information—this figure subtitle was in error and has been edited.

6) Saliva, the material used to obtain genomic DNA, is composed mainly of white blood cells. A hypothesis for explaining the high proportion of somatic mutations identified in P2, a patient treated for colon cancer, is clonal hematopoiesis. Are those somatic mutations enhanced for C>A mutations (COSMIC Sig 18, due to *MUTYH*) or for T>G mutations (~COSMIC Sig 17, due to 5-FU) (Christensen, 2019, Nat Communication - doi.org/10.1038/s41467-019-12594-8)?

This is a great suggestion. We carried out mutation signature fitting using the mutations found in P2 using both COSMIC v2 and v3.3. However, we did not detect the activity of SBS18/36; only signatures SBS1 and SBS5 were assigned during signature decomposition (see figure below).

We then plotted the proportional 1-mer mutation spectrum for P2 and compared it to the 1-mer spectrum expected based on parental ages under the Jonsson et al. parental aging model (see figure below). (Note that comparing expected counts between the spectra would not be appropriate here, as the overall counts in P2 are likely elevated due to somatic mutations.)

While there is a slight increase in A>C (T>G), A>G (T>C), A>T (T>A), and C>A fractions, and a slight decrease in C>G and C>T fractions, these differences were not significant based on a 2x2 contingency table Chi-Square test containing the counts of each mutation type observed in P2 versus expected under the parental aging model:

C>A mutation counts in P2	C>A mutation counts expected under Jonsson parental age model
non-C>A mutation counts in P2	non-C>A mutation counts under Jonsson parental age model

The Chi-Squared test did not result in any p-values <0.05 for any mutation type, indicating that the spectrum of P2 is not significantly different from expectation. We repeated the analysis using Fisher's Exact Test to account for cells with <5 mutations, and the results remained non-significant. While we agree that chemotherapy is a potential explanation for the excess mutation load in P2, we were not able to confidently identify the etiology of this load. We have included these new analyses in a supplementary figure (shown below).

Figure S5. Analysis of spectra in the individual (P2) with elevated number of mutations. A) Comparison of the 1-mer de novo mutation spectrum for individual P2, who showed signs of somatic mutation contamination. The proportional 1-mer mutation spectrum was compared to that expected based on the ages of P2’s parents at the time of their birth under the parental aging model. Each mutation type’s count relative to the sum of the other five types was compared to model expectations using a 2x2 Chi-Squared test and Fisher’s Exact test. No comparison resulted in a p-value < 0.05 (labeled as NS for non-significant), indicating that the spectrum is not significantly different from expectation. Note that these mutations did not go through extensive filtering or manual inspection due to P2 being excluded from further analysis early in the pipeline. The Chi-Squared test was used instead of the Poisson-based test for other individuals because P2’s mutation counts were so highly elevated above the counts expected under the parental aging model, so comparing relative proportions of mutation types was more suitable. **B)** We carried out mutation signature decomposition on the 3-mer spectrum of P2 using SigProfilerExtractor and both the COSMIC v3.3 and v2 catalogs (results using v3.3 shown). The mutation signature was decomposed into SBS signatures 1 and 5; signatures SBS18 and SBS36 did not appear.

7) It seems to be a discrepancy between Figure 4 A and 4B. Figure 4B shows increased C>A mutations only in families “1” and “2” both reporting biallelic mothers. However, Figure 4A, the heatmap showing the observed/expected ratios, C>A mutations in family 4 is very similar (and higher) than family 2. Please explain.

Due to the revision of our mutation calling pipeline, Family 4 no longer has an elevated C>A mutation burden. In the previous version of our data, Family 4 did have an elevated C>A ratio whose observed-over-expected odds ratio was greater than the one in Family 2, but this was due to stochasticity and did not rise to the level of statistical significance. Since family 4 has only two children whose mutations were called using a surrogate father, this family has at most half as much accessible genome for DNM calling compared to Family 1 (which has two children and DNMs called using both parents) and Family 2 (in which DNMs are called using the surrogate method applied to three children).

8) “In each case, the extracted signature was deconvoluted into signatures SBS1 and SBS5, two clock-like signatures that generally make up the bulk of mutations in both germline and somatic data. No contributions of SBS18 or SBS36, somatic mutational signatures associated with defective MUTYH, were detected.”

Consider running SigProfiler using COSMIC version 2 (Alexandrov 2013) instead of version 3 (Alexandrov 2020). MUTYH deficiency was initially associated with Signature 18 (Pilati 2017, Alexandrov is one of the authors – DOI: 10.1002/path.4880). Pilati report MUTYH deficiency in tumors: “Mutational signature analysis identified three distinct signatures closely related to the previously described signatures 1, 5, and 18”.

Another study associated MUTYH deficiency with Sig18 but did not achieve the same correlation using only SBS18 nor SBS36. However, the correlation obtained with Sig18 vs (SBS18+SBS36) was very high (Barreiro, 2022 - DOI: 10.1002/path.5829).

We appreciate this suggestion, and we re-ran SigProfilerExtractor using COSMIC v2. This yielded results that were qualitatively similar to our original results that were obtained using COSMIC v 3.3 (comparison below, with spectra aggregated per family or per individual).

Per-Family Signature Decomposition

Comparison of per-family 3-mer spectra signature decomposition using COSMIC v3.3 (shown in manuscript Figure S14) and COSMIC v2.

Per-Individual Signature Decomposition

Comparison of per-individual 3-mer spectra signature decomposition using COSMIC v3.3 (shown in manuscript Figure S14) and COSMIC v2.

We are likely under-powered to detect the activity of SBS18 in germline data, but do not consider this lack of evidence of activity to rule out the presence of SBS18 if we had

sufficient mutations to more thoroughly explore the triplet mutation spectrum. We have updated the text to reflect this uncertainty in lines 443-446:

However, mutational signature analysis of our DNM data did not identify any activity of either of the MUTYH-associated signatures SBS18 or SBS36. This likely reflects the small total sample size of C>A mutations in our data (**Figure S15**) and should not be interpreted as evidence of absence of SBS18/SBS36.

We have also mentioned the similarity in results in the legend of **Figure S15**.

September 30, 2024

GENETICS-2024-307494

A maternal germline mutator phenotype in a family affected by heritable colorectal cancer

Dear Kelley:

I have now reviewed your paper myself. I am pleased to inform you that, with minor revisions, it is potentially suitable for publication in GENETICS. I do have some comments and concerns that need to be addressed in a revised manuscript, which you can read at the end of this email.

Let me just say that the paper is much improved, and really exciting. I think both the empirical results and new methods will be of interest to lots of people. I hope my comments simply help to clarify the major take-home messages.

We look forward to receiving your revised manuscript. Please let the editorial office know approximately how long you expect to need for revisions.

Upon resubmission, please include:

1. A clean version of your manuscript;
2. A marked version of your manuscript in which you highlight significant revisions carried out in response to the major points raised by the editor/reviewers (track changes is acceptable if preferred);
3. A detailed response to the editor's/reviewers' comments and to the concerns listed above. Please reference line numbers in this response to aid the editors.

Additionally, please ensure that your resubmission is formatted for GENETICS.

<https://academic.oup.com/genetics/pages/general-instructions>

Follow this link to submit the revised manuscript: Link Not Available

Sincerely,

Matthew Hahn
Associate Editor
GENETICS

Approved by:
Amy MacQueen
Senior Editor
GENETICS

AE comments:

1. Mutation notation/language

Throughout, I was a bit confused by the language used to describe MUTYH genotypes. Part of this is certainly the fact that I don't read human medical genetics papers a lot, but the fact that mutation notation (e.g. Y179C) is also used as a heterozygote genotype designation (I think) is very confusing. The same is true for calling individuals "monoallelic" when they are heterozygous at one of the two sites of interest and "biallelic" when they're heterozygous at both (again, I think this is what it means). This is not language that will be familiar to many.

Here are some specific comments about the language and notation as it is first used in the paper:

-line 31: is this level of detail necessary for the Abstract? What does it mean? Is it two mutations? Why are the transcripts reported? At this point it is very unclear.

-Relatedly, on line 35: why is "biallelic" here? Is this important? It's also in quotes on line 81, but not explained there either. (Neither is "monoallelic" on line 86.) Individuals are also referred to as (e.g.) "mothers with biallelic mutations Y179C and V234M", which is confusing.

I think this is all explained on lines 114-116. Note, however, that maybe a "p." is missing from in front of "V234M" on line 115? (I also don't know what the "c." and "p." notation means.)

2. Organization of the Results

The first sub-section describes sequencing, but no results on mutations, as these come three sub-sections later. The second sub-section skips right to experiments in cell lines. Maybe these cell line experiments belong later?

The third sub-section of the Results does describe results around the development of surrogate parents, but no actual "application". Maybe just remove this word from the sub-section title?

There are estimates presented in the next section, but it's not clear which methods those are generated from. Table S1 also doesn't seem to specify, though some of this is specified in the Methods.

3. Explanation and interpretation of surrogate method

When there are two siblings, can we really polarize one of them as the surrogate parent? I think I do understand that the method simply assigns a non-focal sibling as the parent in order to identify "Mendelian violations", but can't we only detect differences between siblings? There does not seem like a way to say which sibling inherited the mutation when they differ in a shared parental haplotype.

Furthermore, what does one do in families with only two siblings? How can we get independent mutation rates for both of siblings, if they have to act as surrogates for each other? What am I missing?

Additional questions about mutations and mutation rates reported:

In Table S1, column K, should we really divide by 2 when using the surrogate parent approach? Isn't only one haplotype (and therefore one haploid genome) being examined?

I'm not sure I understand "phased" mutations in the context of these results. When carrying out surrogate-parent mutation calling, doesn't that define which parent the mutation is inherited from? How was Unfazed used in these cases?

line 311: I'm not sure I understand the title of this sub-section, as well as some of the language used here and later. If P1-P4 are the pathogenic MUTYH carriers, we are not assessing the mutation rates of their children: we are instead assessing their mutation rates by looking in their children. In addition, shouldn't the unusual mutation rate for individual P2 be instead a function of mutations from her parents? She does not have an unusual mutation rate, as evidenced by the rates estimated in her three children (P2 is still used to call mutations for those individuals, correct?). The language in this sub-section should be clarified.

4. Minor comments

-line 71: BER was just defined on line 65. Maybe remove the mention on line 65?

-Please point to individuals P1-P4 when they are first mentioned on lines 113-118 (these are the individuals referenced, correct?).

-line 176: maybe point to Figure 1 here?

-Figure 2A: aren't the lines pointing to shared *maternal* haplotypes?

-lines 200-202: this is a similar confusion as to the one above (referencing line 311) about whose mutation rates we are actually assessing in these types of experiments.

-lines 249-251: this is what I understood to be the "sibling approach" in Jonsson et al. 2018, which they use when gonosomal mutations are present in the parents at low VAFs.

-Figure 3A (red text): missing "c" in inaccessible.

We thank the editor for paying such careful attention to our manuscript and providing these helpful, constructive comments. We have revised the manuscript in response and the revisions are summarized below. In keeping with the formatting requirements of *Genetics*, we have also relocated the Materials and Methods section before the results and shortened it, moving much of the methodological detail to a new Supplementary Methods document.

AE comments:

1. Mutation notation/language

Throughout, I was a bit confused by the language used to describe *MUTYH* genotypes. Part of this is certainly the fact that I don't read human medical genetics papers a lot, but the fact that mutation notation (e.g. Y179C) is also used as a heterozygote genotype designation (I think) is very confusing. The same is true for calling individuals "monoallelic" when they are heterozygous at one of the two sites of interest and "biallelic" when they're heterozygous at both (again, I think this is what it means). This is not language that will be familiar to many.

We agree that this medical genetics language might be confusing to many of the readers we are trying to reach with this manuscript. We have kept this language in the paper since it is standard in the literature on *MUTYH*-associated polyposis, but we have made an effort to define and explain all such terminology for the benefit of readers from evolutionary genetics and other relevant disciplines (lines 87-92):

“The genotypes that cause MAP are often referred to as “biallelic”: biallelic *MUTYH* genotypes are either homozygous for a single pathogenic variant or are compound heterozygotes, with each copy of *MUTYH* affected by a different pathogenic mutation. In contrast, individuals who have “monoallelic” genotypes (meaning heterozygous for a single pathogenic *MUTYH* variant) do not generally develop intestinal polyposis and have at most a modestly elevated cancer risk (Barreiro et al. 2022).”

We have edited the paper to avoid using p.Y179C and p.V234M to refer to monoallelic genotypes, and we now use the more explicit terminology “monoallelic for p.V234M.”

We hope that these notation choices will make the paper accessible to the broad readership of *Genetics* while still maintaining the paper's connection to the literature on MAP and heritable cancer syndromes.

Here are some specific comments about the language and notation as it is first used in the paper:

-line 31: is this level of detail necessary for the Abstract? What does it mean? Is it two mutations? Why are the transcripts reported? At this point it is very unclear.

We have moved these details related to the transcript from the abstract to the main text and added more context about what they mean. The reason for including them, which we hope is now clearer, is that they are needed to interpret the names of the variants, i.e. Y179C and V234M. Much as the coordinates of a SNP depend on the genome assembly being used, the coordinates of an amino acid change depend on the reference transcript being used. For example, since Y179C is a substitution in the 179th residue of the protein, a difference in how the protein is spliced can cause the same DNA substitution to translate to a different amino acid number. Here, we have chosen the named “Y179C” and “V234M” because they are the names of these variants found most commonly in the literature, but unhelpfully, these same variants sometimes go by other names that correspond to other splice variants of the *MUTYH* protein. We hope lines 233-238 now make the definition of this genotype more transparent:

“We obtained saliva samples from three full siblings (labeled P1, P2, and P3 in **Figure 1**) who share a biallelic *MUTYH* genotype known as p.Y179C/V234M. (One gene copy has a tyrosine-to-cysteine substitution at amino acid position 179 and the other has a valine-to-methionine substitution at position 234. Both amino acid positions are indexed in the coordinates of *MUTYH* transcript NM_001128425; substitutions p.Y179C and p.V234M have DNA coordinates c.536A>G and c.700G>A, respectively.) We also sampled saliva from a fourth sibling (P4 in **Figure 1**) who is a monoallelic carrier of p.V234M.”

-Relatedly, on line 35: why is "biallelic" here? Is this important? It's also in quotes on line 81, but not explained there either. (Neither is "monoallelic" on line 86.) Individuals are also referred to as (e.g.) "mothers with biallelic mutations Y179C and V234M", which is confusing. I think this is all explained on lines 114-116. Note, however, that maybe a "p." is missing from in front of "V234M" on line 115? (I also don't know what the "c." and "p." notation means.)

Thank you for flagging the confusion that these terms can cause. We have removed “biallelic” from the abstract since its usage there likely causes more confusion than clarity. Our reason for including it is that *MUTYH* genotypes are generally pathogenic only when both copies of the gene contain pathogenic variants, so a “biallelic” Y179C/V234M genotype is one where the two deleterious mutations are on different copies of the gene, meaning that there is no normal copy of *MUTYH*. In contrast, if both pathogenic variants occurred on the same gene copy and the other gene copy contained no pathogenic variation, that would have very different clinical significance (the person would not be diagnosed with MAP and would have little to no elevation of their cancer risk). However, since these pathogenic variants are rare, they will almost always occur on different copies of the gene, so in practice there is little ambiguity caused by failing to specify that the genotype is biallelic.

We now define monoallelic and biallelic the first time the terms are used in the introduction (lines 87-92 quoted above). We believe we have eliminated all sloppy omissions of the “p.” prefix (which refers to a position in the protein sequence, as opposed to “c.” for gene coordinate). We also agree that the terminology “mother with biallelic *MUTYH* mutations” was confusing, and have replaced it throughout with “mother with a biallelic *MUTYH* genotype” or “mother with the

biallelic genotype p.Y179C/V234M.” We are hopeful that these tidying efforts will make the terminology less painful.

2. Organization of the Results

The first sub-section describes sequencing, but no results on mutations, as these come three sub-sections later. The second sub-section skips right to experiments in cell lines. Maybe these cell line experiments belong later?

Rather than move the cell line experiments later, we decided to combine the first two subsections into a single first section titled “*Whole genome sequencing from a family affected by a MUTYH genotype that shows reduced DNA repair efficiency in vitro,*” which describes the family structure, the functional interrogation of the family’s genotype, and then the sequencing to test our hypothesis about mutator activity. We worried that moving the functional validation to a later point in the manuscript might worry medical genetics readers who might think it’s a big deal that p.V234M is annotated as a variant of uncertain significance in ClinVar and need the functional assay to establish that p.Y179C/V234M is a likely pathogenic genotype. With this restructuring, the sequencing now leads directly into the DNM analysis, which will hopefully be more intuitive.

The third sub-section of the Results does describe results around the development of surrogate parents, but no actual “application”. Maybe just remove this word from the sub-section title?

Good idea, we have revised this title to “*Using siblings as “surrogate parents” for de novo mutation calling in incomplete nuclear families.*”

There are estimates presented in the next section, but it's not clear which methods those are generated from. Table S1 also doesn't seem to specify, though some of this is specified in the Methods.

Good point, we have added a column to Table S1 called “Parents and surrogates used for DNM calling” that now lists the configuration of parents and surrogate parents used for DNM calling in each child.

3. Explanation and interpretation of surrogate method

When there are two siblings, can we really polarize one of them as the surrogate parent? I think I do understand that the method simply assigns a non-focal sibling as the parent in order to identify “Mendelian violations”, but can't we only detect differences between siblings? There does not seem like a way to say which sibling inherited the mutation when they differ in a shared parental haplotype.

The reason that we can polarize mutations called using one or two surrogate parents is that even in standard mutation calling pipelines that use two real parents, mutation calling is only attempted at sites where both parents are homozygous for the reference allele (as described in

the Mutationathon paper by Bergeron, et al. 2022). Since we follow this convention when performing surrogate parent calling as well, each surrogate parent mutation call occurs at a site where one sibling is heterozygous and all other siblings are homozygous reference. It is thus unambiguous that if a mutation occurred at the site, it occurred in the heterozygous sibling. We now explain this in lines 320–324:

“As in standard DNM calling pipelines (Bergeron et al. 2022), we only call mutations at sites where both the real and surrogate parents are homozygous for the reference allele. This allows us to polarize mutation calls with confidence: if we are calling DNMs in Sibling 1 using Sibling 2 as a surrogate parent, all DNM calls will occur at sites where Sibling 1 is heterozygous and Sibling 2 is homozygous, which are not variants that could also be explained by DNMs in Sibling 2.”

Furthermore, what does one do in families with only two siblings? How can we get independent mutation rates for both of siblings, if they have to act as surrogates for each other? What am I missing?

We tried to tailor the new language in lines 320–324 (quoted above) to make it clear how one can obtain two independent mutation rate estimates with just 2 siblings. If we let S1, S2, and M be two siblings and their mother, DNMs in S1 will have the genotype configuration S1: 0/1; S2: 0/0, M: 0/0. In contrast, DNMs in S2 will have the genotype configuration S1:0/0, S2:0/1, M:0/0. As we note in the manuscript, we are not able to detect DNMs that are shared among siblings, but for the vast majority of human mutations that are private to one sibling, we can tell unambiguously which sibling they occurred in because only that sibling will be heterozygous.

Additional questions about mutations and mutation rates reported:

In Table S1, column K, should we really divide by 2 when using the surrogate parent approach? Isn't only one haplotype (and therefore one haploid genome) being examined?

It should still be appropriate to divide by 2 in this case because in each region where mutations are callable using a surrogate parent approach we are able to call mutations that originated on both the maternal and paternal haplotypes. In order to distinguish DNMs from inherited mutations, we always need to be able to compare the proband's genome to some individual who shares their maternal haplotype (either their mother or a surrogate mother) plus some individual who shares their paternal haplotype (either their father or a surrogate father). Given any sibling surrogate father, the most recent common ancestor of the surrogate father and the proband is the real father, so we will still be able to identify sites where the proband's genome differed from the real father's genome, since the real father's genotype was transmitted unmutated to the surrogate father.

To clarify this point as well as the following point regarding phasing, we have added the following paragraph (lines 367-376):

“Within each region that is accessible for DNM calling, most DNMs occurring on the proband’s maternal and paternal haplotypes should be identifiable, with the exception of DNMs that are shared with the sibling surrogate parent. Neglecting these sib-shared mutations, the mutation rate can be estimated by dividing the mutation count by two times the length of the accessible genomic region spanned by paternal IBD tracts. Moreover, read-backed phasing tools that are designed for use in parent/child trios can be applied with the surrogate father substituted for the father (Belyeu et al. 2021). Read-backed phasing will deduce that a mutation arose on a paternally inherited chromosome if it can be phased to a haplotype shared between the siblings that is not shared with their mother. Similarly, we can deduce that a mutation occurred on a maternally inherited chromosome if it can be phased to a haplotype that is shared with the maternal genome.”

I'm not sure I understand "phased" mutations in the context of these results. When carrying out surrogate-parent mutation calling, doesn't that define which parent the mutation is inherited from? How was Unfazed used in these cases?

As we attempted to clarify above, we actually can use the surrogate parent approach to call both paternal and maternal mutations. We can use Unfazed in much the same way that it is used when both parental genomes are available: if a mutation occurs on a read containing an inherited variant that is present in the surrogate father but not in the mother, we can infer that the DNM arose on the haplotype the proband inherited from their father (which was also inherited by the surrogate father in the absence of the DNM).

It is true that when we use two surrogate parents, phasing starts to lose meaning, since we are usually not sure which sibling is acting as the surrogate father and which sibling is acting as the surrogate mother. This is why the phasing-related columns in Table S1 (M through P) are labeled “NA” in P1–P4.

line 311: I'm not sure I understand the title of this sub-section, as well as some of the language used here and later. If P1-P4 are the pathogenic MUTYH carriers, we are not assessing the mutation rates of their children: we are instead assessing their mutation rates by looking in their children. In addition, shouldn't the unusual mutation rate for individual P2 be instead a function of mutations from her parents? She does not have an unusual mutation rate, as evidenced by the rates estimated in her three children (P2 is still used to call mutations for those individuals, correct?). The language in this sub-section should be clarified.

This is a good point, we have gone through the manuscript and eliminated all references to the mutation rate of a child, referring instead to the mutation rate of a parent/child trio. In some cases, we still refer to a child’s mutation “load,” which seems appropriate since the child’s genome is where the mutations ultimately end up.

P2 is a bit of a different case since the low allelic balance of her excess mutations leads us to conclude that most of these are somatic mutations that arose in her own hematopoietic cell lineage (and indeed, the evidence from her children suggests that her germline is not affected by this high mutation rate). We have rephrased the description of P2 as a mutation frequency

rather than a rate to more precisely describe the fact that we found a lot of mutations in this genome (lines 465-466):

“However, we called a higher frequency of mutations (4.4×10^{-8} mutations per site per generation) in the genome of P2, the biallelic mother of Family 2.”

4. Minor comments

-line 71: BER was just defined on line 65. Maybe remove the mention on line 65?

Done, thanks for catching this.

-Please point to individuals P1-P4 when they are first mentioned on lines 113-118 (these are the individuals referenced, correct?).

Done, good suggestion.

-line 176: maybe point to Figure 1 here?

Done.

-Figure 2A: aren't the lines pointing to shared *maternal* haplotypes?

Yes good catch, this is now fixed.

-lines 200-202: this is a similar confusion as to the one above (referencing line 311) about whose mutation rates we are actually assessing in these types of experiments.

We have rephrased this from a statement about an “individual’s” mutation rate to a statement about a “trio’s” mutation rate, which we hope will remove this ambiguity (lines 331-332 of the revised manuscript):

“... our method is designed to estimate an individual parent/child trio’s mutation rate using mutations that arose within a single generation.”

-lines 249-251: this is what I understood to be the "sibling approach" in Jonsson et al. 2018, which they use when gonosomal mutations are present in the parents at low VAFs.

Thanks for the clarification, we had overlooked this part of Jonsson, et al. 2018 and we now cite this analysis in lines 316–318:

“Jónsson, et al. (2018) previously used a similar procedure to identify mutations that arose early in parental embryonic development, which often have parental read support and are thus not detectable by standard parent/child trio mutation calling methodology.”

-Figure 3A (red text): missing "c" in inaccessible.

Thanks for catching this, it has been fixed.

October 7, 2024

RE: GENETICS-2024-307519

Dr. Kelley Harris
University of Washington
Genome Sciences
3720 William H Foege Hall
Seattle, Washington 98105

Dear Dr. Harris:

Congratulations! We are delighted to inform you that your manuscript entitled "A maternal germline mutator phenotype in a family affected by heritable colorectal cancer" is acceptable for publication in GENETICS. Many thanks for submitting your research to the journal.

To Proceed to Production:

1. Format your article according to GENETICS style, as discussed at <https://academic.oup.com/genetics/pages/general-instructions>, and upload your final files at <https://genetics.msubmit.net>.
2. Your manuscript will be published as-is (unedited-as submitted, reviewed, and accepted) at the GENETICS website as an Advanced Access article and deposited into PubMed shortly after receipt of source files and the completed license to publish. Please notify sourcefiles@thegsajournals.org if you do not wish to publish your article via Advanced Access.
3. We invite you to submit an original color figure related to your paper for consideration as cover art. Please email your submission to the editorial office or upload it with your final files. You can submit a small-sized image for evaluation, and if selected, the final image must be a TIFF file 2513px wide by 3263px high (8.375 by 10.875 inches; resolution of 600ppi). Please avoid graphs and small type.

If you have any questions or encounter any problems while uploading your accepted manuscript files, please email the editorial office at sourcefiles@thegsajournals.org.

Sincerely,

Matthew Hahn
Associate Editor
GENETICS

Approved by:
Amy MacQueen
Senior Editor
GENETICS

note: Please add jnls.author.support@oup.com and genetics.oup@kwgglobal.com (or the domains @oup.com and @kwgglobal.com) to your email program's "safe senders" list. You will be contacted by both at various points during the production process.